# Structural dynamics of RbmA governs plasticity of *Vibrio cholerae* biofilms

**Jiunn CN Fong[1], Andrew Rogers[1], Alicia K Michael[2], Nicole C Parsley[2†], William-Cole Cornell[3], Yu-Cheng Lin[3], Praveen K Singh[4], Raimo Hartmann[4], Knut Drescher[4], Evgeny Vinogradov[5], Lars EP Dietrich[3], Carrie L Partch[2]\*, Fitnat H Yildiz[1]\***

[1]Department of Microbiology and Environmental Toxicology, University of California, Santa Cruz, Santa Cruz, United States; [2]Department of Chemistry and Biochemistry, University of California, Santa Cruz, Santa Cruz, United States; [3]Department of Biological Sciences, Columbia University, New York, United States; [4]Max Planck Institute for Terrestrial Microbiology, Marburg, Germany; [5]National Research Council, Ottawa, Canada

**Abstract** Biofilm formation is critical for the infection cycle of *Vibrio cholerae*. *Vibrio* exopolysaccharides (VPS) and the matrix proteins RbmA, Bap1 and RbmC are required for the development of biofilm architecture. We demonstrate that RbmA binds VPS directly and uses a binary structural switch within its first fibronectin type III (FnIII-1) domain to control RbmA structural dynamics and the formation of VPS-dependent higher-order structures. The structural switch in FnIII-1 regulates interactions in trans with the FnIII-2 domain, leading to open (monomeric) or closed (dimeric) interfaces. The ability of RbmA to switch between open and closed states is important for *V. cholerae* biofilm formation, as RbmA variants with switches that are locked in either of the two states lead to biofilms with altered architecture and structural integrity.
DOI: https://doi.org/10.7554/eLife.26163.001

**\*For correspondence:** cpartch@ucsc.edu (CLP); fyildiz@ucsc.edu (FHY)

**Present address:** †Department of Chemistry, University of North Carolina, Chapel Hill, United States

**Competing interests:** The authors declare that no competing interests exist.

## Introduction

In nature, microorganisms are predominantly found in biofilms, embedded within a matrix composed of exopolysaccharides, proteins, lipids and nucleic acids. Matrix components provide structural integrity, facilitate cell-cell and cell-surface attachment and enhance population fitness (*Flemming and Wingender, 2010*; *Nadell et al., 2016*). The molecular underpinnings of matrix assembly, however, is not fully understood.

Biofilm formation is critical for both aquatic and intestinal phases in the lifecycle of the facultative human pathogen *Vibrio cholerae,* the causative agent of the severe diarrheal disease cholera (*Teschler et al., 2015*). The ability to form biofilms enhances growth and survival of the pathogen outside the body of its mammalian host by providing protection from a number of environmental stresses, including nutrient limitation and predation by protozoa and bacteriophages (*Yildiz and Schoolnik, 1999*; *Matz et al., 2005*; *Beyhan and Yildiz, 2007*). Additionally, growth in biofilms enhances transmission and induces a hyper-infectious phenotype (*Faruque et al., 2006*). Production of biofilms by *V. cholerae* requires extracellular matrix components, such as the *Vibrio* exopolysaccharide (VPS) and the matrix proteins RbmA, RbmC, and Bap1 (*Yildiz and Schoolnik, 1999*; *Fong et al., 2006*; *Fong and Yildiz, 2007*; *Fong et al., 2010*; *Absalon et al., 2011*; *Berk et al., 2012*). RbmA is thought to capture progeny to the founder cell, thereby enabling cell-cell adhesion and formation of cell clusters (*Berk et al., 2012*). Additionally, RbmA controls the arrangements of *V. cholerae* cells relative to the substratum during biofilm development, and in turn, governs development of mature biofilm architecture (*Drescher et al., 2016*).

We previously determined the crystal structure of RbmA, which showed that RbmA is composed of tandem fibronectin type III (FnIII) domains (*Giglio et al., 2013*). However, we do not yet understand the molecular mechanisms by which RbmA contributes to biofilm formation. To determine how *V. cholerae* biofilm matrix is built and how biofilms attain their structural integrity, we further characterized RbmA structure and analyzed the RbmA-VPS interaction. Here, we show that RbmA binds VPS directly to control the VPS-dependent formation of higher-order structures that modulate biofilm architecture and that *V. cholerae* biofilms can be fine-tuned by RbmA structural dynamics.

## Results

### RbmA undergoes a binary structural switch

Structural analysis of RbmA revealed that two tandem FnIII domains (FnIII-1 and FnIII-2), which are connected by a short linker and interact in trans in an antiparallel orientation to form a stable dimer (*Giglio et al., 2013*; *Maestre-Reyna et al., 2013*). The native RbmA dimer is characterized by two distinct structures at the dimer interface in multiple crystal forms, where residues 91 to 108 of FnIII-1 adopt so-called disordered-loop (D-loop, *Figure 1A*, top) or ordered-loop (O-loop, *Figure 1A*, bottom) conformations (*Giglio et al., 2013*; *Maestre-Reyna et al., 2013*). In the O-loop conformation, the βc strand of the FnIII-1 domain is largely unfolded to facilitate a network of interactions between residue D97 and the FnIII-1 and FnIII-2 domains to stabilize the closed dimer interface, whereas these interactions are abolished in the D-loop conformation (*Figure 1B* and *Figure 1—figure supplement 1*). One key residue interacting with D97 in the O-loop conformation is FnIII-2 residue R234. We previously showed that the R234A mutant disrupted biofilm formation (*Giglio et al., 2013*), suggesting that regulation of this dimer interface might be important for RbmA function.

To explore how these two discrete conformations at the dimer interface impact RbmA structure and thereby biofilm formation, we first generated RbmA variants carrying a D97A or D97K mutation with the goal of disrupting O-loop interactions. We then compared $^{15}$N-$^{1}$H TROSY-HSQC NMR spectra of uniformly $^{15}$N-labeled D97 mutants to wild-type (WT) and R234A RbmA. The overlaid spectra revealed several residues in the FnIII-2 domain, which were distal from any mutation sites that exhibited collinear chemical shifts reporting on a two-state equilibrium in fast exchange, with populations of the two states tuned by different mutations (*Figure 1C* and *Figure 1—figure supplement 2*). The endpoints of this two-state equilibrium are punctuated by the R234A and D97A or D97K mutants, suggesting that they represent discrete structural states of the switch. Each peak for the WT protein was located approximately halfway between the two endpoints, consistent with observations that the protein natively populates both states (*Giglio et al., 2013*; *Maestre-Reyna et al., 2013*).

To identify which of the two endpoints correspond to the O-loop and D-loop states, we also analyzed the truncated, recombinant RbmA* protein, which corresponds to the predicted product of a regulated proteolytic event in vivo that disrupts the FnIII-1 domain (*Smith et al., 2015*). Comparing the $^{15}$N-$^{1}$H TROSY-HSQC spectrum of $^{15}$N RbmA* to the full-length variants, we observed a striking similarity of the chemical shifts for the full-length D97 mutants and those of RbmA*, indicating that mutations at D97 lock RbmA into the open, D-loop conformation (*Figure 1C* and *Figure 1—figure supplement 2*). Based on the two states consistently observed in RbmA crystal structures (*Giglio et al., 2013*; *Maestre-Reyna et al., 2013*), this implies that the R234A mutation locks RbmA into the more closed O-loop conformation.

To determine how changes in this binary switch in the FnIII-1 domain might influence RbmA dimerization state, we performed size exclusion chromatography in-line with multi-angle light scattering (SEC-MALS) on our series of purified RbmA variants (*Figure 1D* and *Table 1*). Consistent with a model where mutations at D97 eliminate O-loop interactions that stabilize the FnIII-1:FnIII-2 dimer interface, both D97A and D97K mutants were predominantly monomeric, while WT and R234A formed stable dimers under these conditions. RbmA* was also found predominantly in its expected monomeric state due to the absence of an intact FnIII-1 domain, although a small fraction (<10%) exhibited higher-order aggregation, likely due to retention of the unfolded FnIII-1 fragment (*Figure 1D*). These data demonstrate that the O-loop is required to stabilize the FnIII-1:FnIII-2 dimer interface, and that formation of only one O-loop interface in the WT protein is sufficient to stabilize the RbmA dimer.

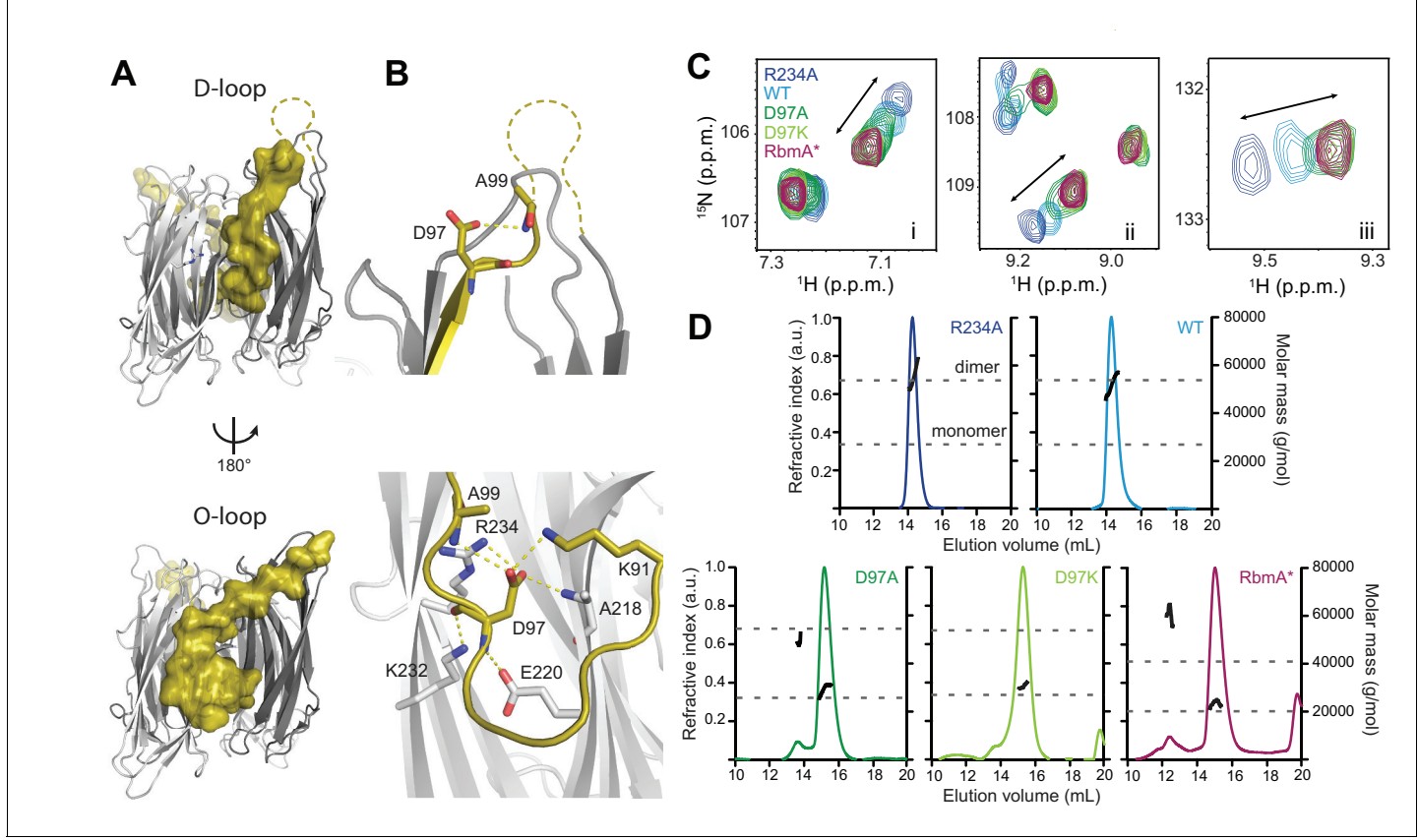

**Figure 1.** RbmA binary switch regulates dimerization. (**A**) Two views of the RbmA dimer structure (PDB: 4KKP) with the FnIII-1 domain binary switch depicted in its two states (D-loop, O-loop) via surface representation (olive). Dashed line, backbone density missing in crystal structure. (**B**) Zoomed-in view of residue D97 in the two states of the switch. Residues that make polar contacts to D97 are highlighted. (**C**) Selected regions of overlaid $^1$H-$^{15}$N TROSY-HSQC spectra of full-length $^{15}$N RbmA proteins illustrating changes in the two-state equilibrium between wild-type (WT) and mutant proteins. Panels i-iii represent portions of the TROSY-HSQC spectra; FnIII-2 domain residues exhibiting collinear shifts are as follows: (i) T180, (ii) G163, and (iii) S225. More information can be found in **Figure 1—figure supplement 2**. (**D**) Analysis of RbmA dimerization state by SEC-MALS. Dashed lines, calculated masses for monomer and dimer masses as indicated.

DOI: https://doi.org/10.7554/eLife.26163.002

The following figure supplements are available for figure 1:

**Figure supplement 1.** Larger view of βc switch region.
DOI: https://doi.org/10.7554/eLife.26163.003

**Figure supplement 2.** Mutations at D97 and R234 influence populations of the binary switch in RbmA.
DOI: https://doi.org/10.7554/eLife.26163.004

To further explore how the FnIII-1 switch regulates global structural differences in RbmA, we performed limited proteolysis on R234A (O-loop conformation), WT (O-loop and D-loop), and D97A (D-loop conformation) to probe for differential sensitivity to trypsin. We reasoned that the two locked

**Table 1.** Size exclusion chromatography in-line with multi-angle light scattering (SEC-MALS) analysis.

| Construct | Calculated MW (kDa) (monomer: dimer) | Observed MW (kDa) |
|---|---|---|
| WT | 26.9: 53.8 | 1 peak; 51.8 ± 0.5% |
| R234A | 26.9: 53.8 | 1 peak; 55.2 ± 0.8% |
| D97A | 26.9: 53.8 | 1 peaks; 33.8 ± 1.2%, 46.6 ± 8.4% |
| D97K | 26.9: 53.8 | 1 peak; 30.6 ± 7.9% |
| RbmA* | 22.0: 44.0 | 2 peaks; 25.7 ± 7.9%, 59.8 ± 20.8% |

DOI: https://doi.org/10.7554/eLife.26163.005

mutants (R234A and D97A) might have signature proteolytic sensitivities that would both be present in the WT protein due to its sampling of the two discrete states. The monomeric D97A mutant was rapidly proteolyzed down to a stable 13 kDa fragment comprising the FnIII-2 domain that was also observed to a lesser degree in the WT protein, but not in the R234A mutant (*Figure 2A–C*). The dimeric WT and R234A proteins both remained relatively stable to proteolysis and shared a higher molecular weight tryptic fragment. The relative protease-resistance of the R234A mutant in vitro is further evidence in that it adopts a closed conformation relative to the D97A/D97K mutants.

RbmA undergoes a proteolytic cleavage in vivo (*Berk et al., 2012*; *Smith et al., 2015*). We found that RbmA was readily cleaved in vivo in both planktonic and biofilm cultures (*Figure 2D and E*). To determine how changes to this binary switch in the FnIII-1 domain might influence RbmA proteolytic cleavage, we analyzed in vivo proteolysis of RbmA in the D97A, D97K and R234A variants and found that they produced the expected proteolytic products (*Figure 2F*). As a control, we included a strain lacking all three proteases (HapA, PrtV and IvaP) reported to cleave RbmA and showed that RbmA proteolysis is markedly decreased in this strain. In a strain lacking VPS, RbmA is completely degraded. As expected, RbmA proteolysis is decreased in the rugose variant (*Figure 2D*), which is characterized by increased VPS production relative to the wild-type smooth strain (*Figure 2—figure supplement 1*). As expected, RbmA proteolysis is decreased in rugose variant (*Figure 2D*), which are characterized by high VPS production, relative to wild-type smooth strain (*Figure 2—figure supplement 1*). Collectively, these data support a model in which RbmA natively samples both the O-loop and D-loop states to regulate dimer interfaces and protease accessibility in vivo.

## RbmA uses a binary structural switch to govern biofilm architecture

To analyze consequences of this binary switch on regulation of biofilm formation by RbmA, we generated mutated versions of *rbmA* (D97A, D97K or R234A) in the chromosome. For these studies, we used a rugose variant of a clinical *V. cholerae* O1 El Tor strain, as this strain produces ample matrix components (*Yildiz and Schoolnik, 1999*; *Yildiz et al., 2014*). Production and secretion of RbmA harboring the mutations was similar to that of the parental strain (*Figure 3—figure supplement 1*).

To analyze overall biofilm architecture, we first compared structures of biofilms formed at solid-air interfaces by examining differences in corrugation patterns of colony biofilms. Strains harboring the D97A and D97K mutations retained the colony corrugation pattern of the rugose parental strain, while strains lacking RbmA (Δ*rbmA*) or harboring the R234A mutation exhibited a marked decrease in colony corrugation (*Figure 3A*). The strains also varied in the colony corrugation development as a function of time, with Δ*rbmA* and R234A showing uniform spreading, D97A and D97K showing irregular spreading, and the parental rugose strain showing an intermediate spreading phenotype (*Figure 3B*, *Figure 3—figure supplement 2F*, and *Video 1*). In dark-field colony images of thin sections prepared at 48 hr, the rugose, D97A, and D97K strains all showed similar vertical wave-like features of biomass distribution. In contrast, the internal colony structures of the Δ*rbmA* and R234A strains were more uniform (*Figure 3B*). Although colony heights were similar among all of the strains tested (*Figure 3—figure supplement 2E*), colony outgrowth of D97 mutants was reduced when compared to the parental rugose strain. Thus, colony biofilms of the rugose parental strain exhibit features observed in mutants representative of both the O-loop and D-loop states of RbmA, with uniform outgrowth akin to that observed for the locked O-loop mutant (R234A) and internal features reminiscent of those observed in the locked D-loop mutants (D97A, D97K).

We then analyzed micro-colony and mature biofilm formation at the solid-liquid interface using flow-cells. Both Δ*rbmA* and R234A strains formed micro-colonies with edges that are more smooth and regularly-shaped as opposed to the irregularly-shaped micro-colonies observed in the parental strain (*Figure 3C*). Quantitative and microscopic analysis of mature biofilms formed by Δ*rbmA* and R234A revealed that they were less structured and exhibited a decrease in biomass and thickness; the biofilm properties of D97A/D97K mutants more closely resembled that of the parental strain (*Figure 3D* and *Figure 3—figure supplement 2A–C*). Biofilms of R234A and Δ*rbmA* strains also exhibited increased cell detachment, resulting in higher colony forming units (CFU) per effluent volume when compared to the parental strain and D97A/D97K mutants (*Figure 3—figure supplement 2D*).

Analysis of biofilm development at single-cell resolution revealed additional differences between the strains with locked FnIII-1 loops. Nearest-neighbor distance, which is a measure of average cell-cell spacing (*Figure 4A* and *Figure 4—figure supplement 1* and *Video 2*), is increased in the

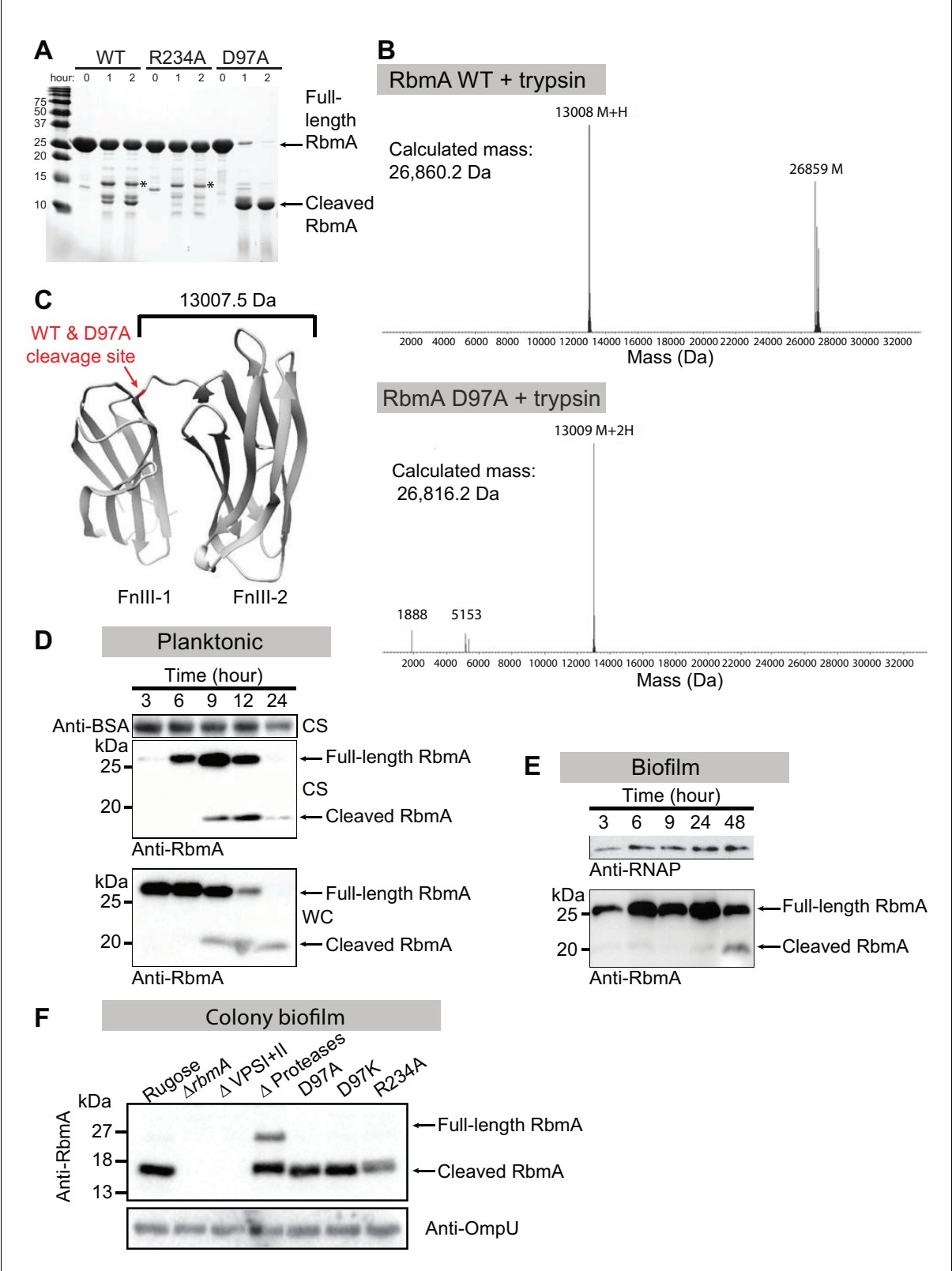

**Figure 2.** Switch mutations at D97 and R234 exert changes in the global structure of RbmA. (**A**) Limited proteolysis of full-length wild-type (WT), R234A and D97A mutants in the presence of 1:150 (w/w) ratio of trypsin for the indicated times at room temperature. Tryptic fragments were resolved by 18% SDS-PAGE and visualized by Coomassie stain. Arrows indicate full-length and the predominant cleaved species that is conserved between RbmA WT and D97A, while asterisks (*) indicate the higher molecular weight fragment that is conserved between WT and R234A. (**B**) Deconvoluted mass

*Figure 2 continued on next page*

Figure 2 continued

spectrometry for tryptic digests of WT and D97A proteins, showing the predominant cleaved species in RbmA WT (~13 kDa and ~26.8 kDa fragments) and D97A (~13 kDa fragment). (C) Location of the cleavage site (red) conserved between WT and D97A proteins mapped onto the RbmA dimer structure (PDB: 4KKP). Western blot analysis of RbmA stability in (D) rugose planktonic and (E) flowcell biofilm cells at various growth phases, and (F) colony biofilm cells after 24 hr of growth. Antibodies against bovine serum albumin (anti-BSA), RNA polymerase (anti-RNAP) and outer membrane protein U (anti-OmpU) were used as additional loading controls. Culture supernatant (CS) fraction fractions were spiked with BSA. WC, whole-cell fraction.

DOI: https://doi.org/10.7554/eLife.26163.006

The following figure supplement is available for figure 2:

**Figure supplement 1.** Western blot analysis of RbmA stability in wild-type smooth strain (low-VPS producing strain) at various growth phases.

DOI: https://doi.org/10.7554/eLife.26163.007

following order: rugose, D97K, D97A, ΔrbmA, R234A. Examination of internal cell arrangements in the biofilm revealed that cells in direct contact with the glass substratum are vertically oriented in the R234A mutant (*Figure 4B* and *Figure 4—figure supplement 1*). Such vertical orientation of cells in contact with the substratum is also displayed by the ΔrbmA strain, albeit to a lesser extent. By contrast, cells of the rugose strain that are in contact with the substratum are mostly radially aligned and horizontal. The D97 mutant strains seem to broadly have similar cell orientations, which are different from both the rugose and the R234A strains. With respect to local nematic order, a measure of the degree of local alignment of the cells (see definition in the Materials and Methods), R234A exhibits a nearly crystal-like cellular alignment in the center and at the bottom of the biofilm (*Figure 4B*). By contrast, decreased nematic order with the D-loop locked mutants and wild-type RbmA strain suggests that sampling the O-loop conformation leads to decreases in local cell alignment.

To determine if the differences in biofilm architectural properties of strains with RbmA variants are due to alterations in localization of RbmA, we determined the RbmA distribution during biofilm formation. For rugose, D97A, and D97K strains, the RbmA localization was similar, where RbmA is distributed between the surface of the biofilm and the center of the biofilm (*Figure 4C*). In contrast, in the R234A strain, RbmA abundance was high at the center of the biofilm, and decreased towards the surface.

Collectively, these data demonstrate that changes in the conformation and/or dimerization state of RbmA alter biofilm architectural properties. Locking RbmA in the closed, O-loop dimer state with the R234A mutation severely affects biofilm formation, while the locked, monomer D-loop state (D97A/D97K) more closely resembles the native role of RbmA in biofilm regulation and that differences in biofilm architecture could be due to changes in RbmA localization.

## RbmA binds VPS via FnIII-2 domain and forms higher-order structures

Based on the extensive role of RbmA in biofilm structure and function and noted requirement of VPS for accumulation of RbmA in the biofilm matrix (*Berk et al., 2012*; *Nadell et al., 2015*), we hypothesized that RbmA might govern biofilm structural and functional properties by interacting with VPS. To first explore RbmA-VPS interactions, we collected $^{15}$N-$^{1}$H TROSY-HSQC spectra in the presence of increasing concentrations of native VPS isolated from *V. cholerae* and monitored chemical shift perturbations in full-length $^{15}$N RbmA. We found that addition of VPS induced uniform peak broadening in $^{15}$N RbmA spectra, consistent with formation of higher-order soluble oligomers (*Figure 5A*). Similar changes in peak intensity were also observed in VPS binding experiments with $^{15}$N RbmA R234A, suggesting that remarkable functional differences between WT and R234A RbmA might not lie in its ability to bind VPS. To determine if perturbation of the FnIII-1 switch influenced formation of higher-order soluble aggregates with VPS, we collected 1D $^{1}$H NMR spectra of full-length WT, R234A, D97A, and the truncated RbmA* protein in the presence of increasing VPS. Integrating peak intensity from either the amide region (11 to 5.5 p.p.m.) (*Figure 5B* and *Figure 5—figure supplement 1*) or side chain region (2.5 to −1 p.p.m.) (*Figure 5C* and *Figure 5—figure supplement 1*) of the $^{1}$H spectra demonstrated that the extent of peak broadening correlates with formation of the locked species. Based on these data, we suggest that binding to VPS by RbmA results in the formation of higher-order RbmA-VPS structures that are governed by the structural dynamics of the binary switch in RbmA.

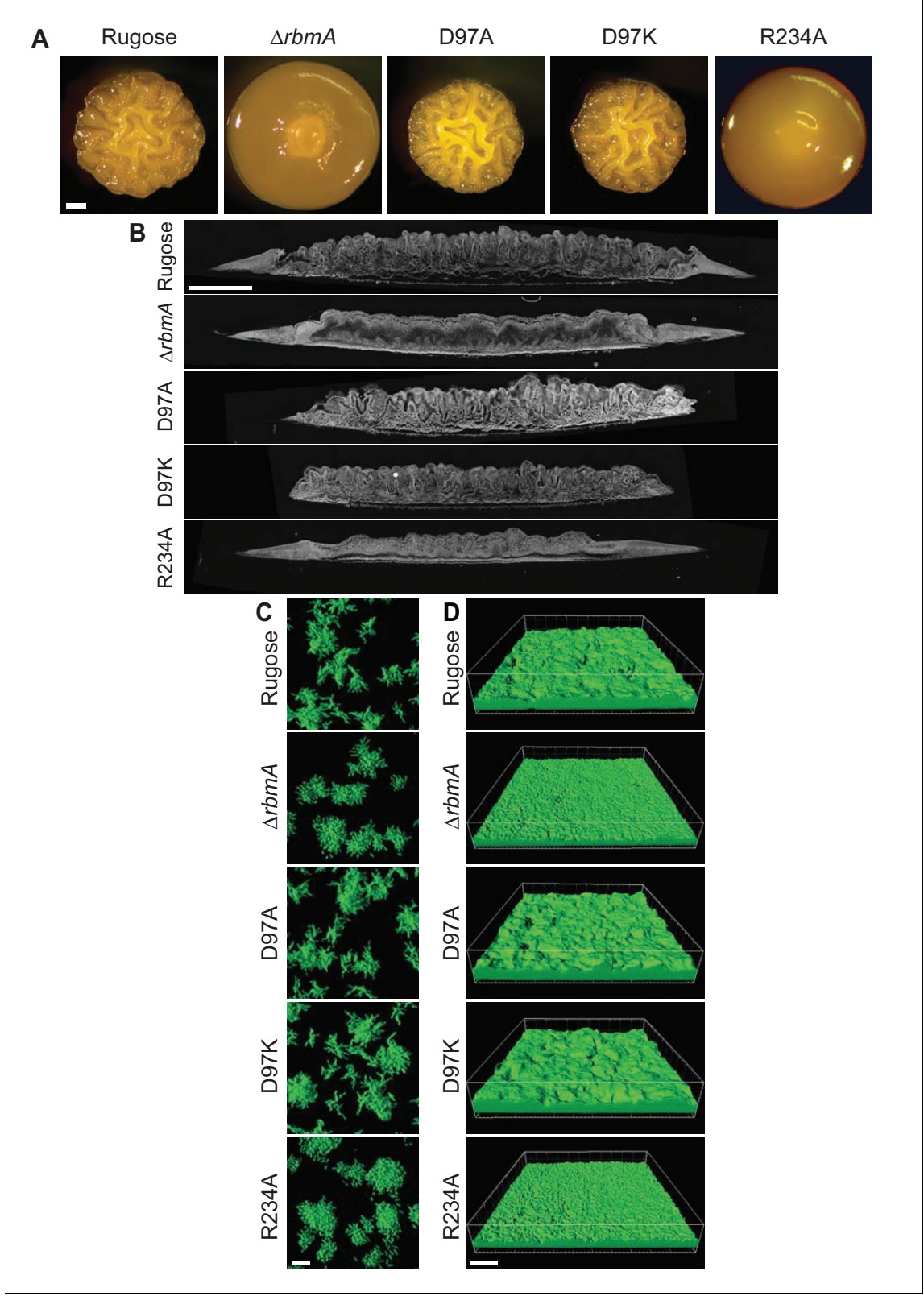

**Figure 3.** RbmA loop conformational transitions impacts biofilm architecture. (**A**) Colony corrugation phenotypes and (**B**) darkfield images of thin-sectioned paraffin-embedded spot colonies after 5 and 2 days of growth, respectively, at 25°C. Scale bars, 0.5 mm (in **A**), 1 mm (in **B**). (**C**) Cell-packing and (**D**) mature biofilm phenotypes of rugose strain, Δ*rbmA* and various mutant strains at 6 hr and 24 hr, respectively. Scale bars, 10 μm (in **C**), 40 μm (in **D**).

*Figure 3 continued on next page*

*Figure 3 continued*

DOI: https://doi.org/10.7554/eLife.26163.008

The following figure supplements are available for figure 3:

**Figure supplement 1.** Analysis of RbmA production and degradation.

DOI: https://doi.org/10.7554/eLife.26163.009

**Figure supplement 2.** Quantitative analysis of (A) biofilm biomass, (B) average biofilm thickness, (C) maximum biofilm thickness, and (D) colony counts (CFU/mL) in the effluent of the biofilms formed by the rugose strain, Δ*rbmA* and various mutant strains at 24 hr post-inoculation as determined by COMSTAT and dilution plating analysis.

DOI: https://doi.org/10.7554/eLife.26163.010

To investigate how VPS is organized in biofilms of rugose, D97A, D97K, and R234A strains, we analyzed VPS localization with FITC-conjugated wheat-germ agglutinin (WGA) and concanavalin A (ConA). We found that VPS is localized around cell clusters in the rugose, D97A and D97K strains (*Figure 5D*). However, in the R234A strain, VPS localization was diffuse and reduced. These findings suggest that structural properties of RbmA affects VPS localization in biofilms.

To begin to explore molecular basis of VPS binding by RbmA, we first studied the isolated $^{15}$N-labeled FnIII domains and RbmA* by NMR spectroscopy. We observed that the full-length protein can undergo spontaneous cleavage in the FnIII-1 domain near the interdomain linker (*Figure 6A* and *Figure 6—figure supplement 1A–C*), resulting in a highly stable 13 kDa protein. Importantly, the $^{15}$N-$^1$H HSQC spectrum of the cleaved form closely resembles the RbmA* variant, although it lacked the poorly dispersed peaks in the center of the spectrum (8.0–8.5 p.p.m.) that correspond to the unfolded FnIII-1 fragment remaining in RbmA* (*Figure 6—figure supplement 1D–I*). Well-resolved peaks in the RbmA* spectrum also exhibited marked similarity to the isolated $^{15}$N FnIII-2 domain. An $^{15}$N-$^1$H HSQC spectrum of the isolated $^{15}$N FnIII-1 domain exhibited poor chemical shift dispersion and a tendency to form soluble aggregates, indicating that this domain is not stably folded in isolation. Altogether, these data suggest that the FnIII-2 domain is the stable core domain of RbmA and its natively processed form, RbmA*, possibly regulates the structural integrity of the FnIII-1 domain through formation of the dimer.

Because RbmA* largely retains RbmA function and has been reported to bind to VPS-containing cells in vivo (*Smith et al., 2015*), we examined whether this truncated form binds VPS directly. We collected $^{15}$N-$^1$H HSQC spectra of $^{15}$N RbmA* in the presence and absence of VPS, observing a number of dose-dependent chemical shift perturbations consistent with VPS binding; the same chemical shifts were also found upon titration of VPS into the isolated $^{15}$N FnIII-2 domain (*Figure 6B*). Chemical shift assignment of the isolated FnIII-2 domain allowed us to identify residues throughout the domain with significant VPS-dependent chemical shift perturbations (*Figure 6C*). When mapped onto the crystal structure of the RbmA dimer, these residues form one contiguous patch on the exposed outer surface of the FnIII-2 domain that likely constitutes the VPS binding site (*Figure 6D*). This interface is distinct from that used to form the RbmA dimer through interaction with the FnIII-1 domain. The presence of two such motifs on each RbmA dimer therefore suggests the possibility for multivalent binding to VPS. Moreover, solution studies of RbmA have demonstrated that the two halves of the dimer, separated by flexible interdomain linkers, can open to form an extended conformation (*Figure 6E*), suggesting that the VPS-binding interface could regulate RbmA function by possibly contributing to the formation of different conformations in solution.

To determine if R234 plays a direct role in VPS binding, we carried out NMR studies using the isolated $^{15}$N FnIII-2 domain harboring the R234A mutation. Introduction of the mutation induces chemical shift perturbations that are localized primarily at the mutation site on the side of the FnIII-2 domain, which does not overlap with the putative VPS binding site (*Figure 6—figure supplement 2*). Correspondingly, we found minimal effect of the R234A mutation on VPS binding by the full-length proteins in our NMR titration studies (*Figure 5A–C*), indicating that R234 does not appear to directly influence VPS binding.

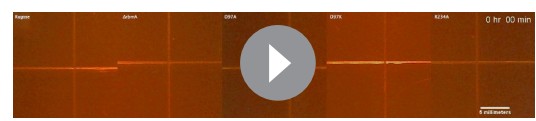

**Video 1.** Time lapse movie of colony biofilm development of rugose strain and various *rbmA* mutants.

DOI: https://doi.org/10.7554/eLife.26163.011

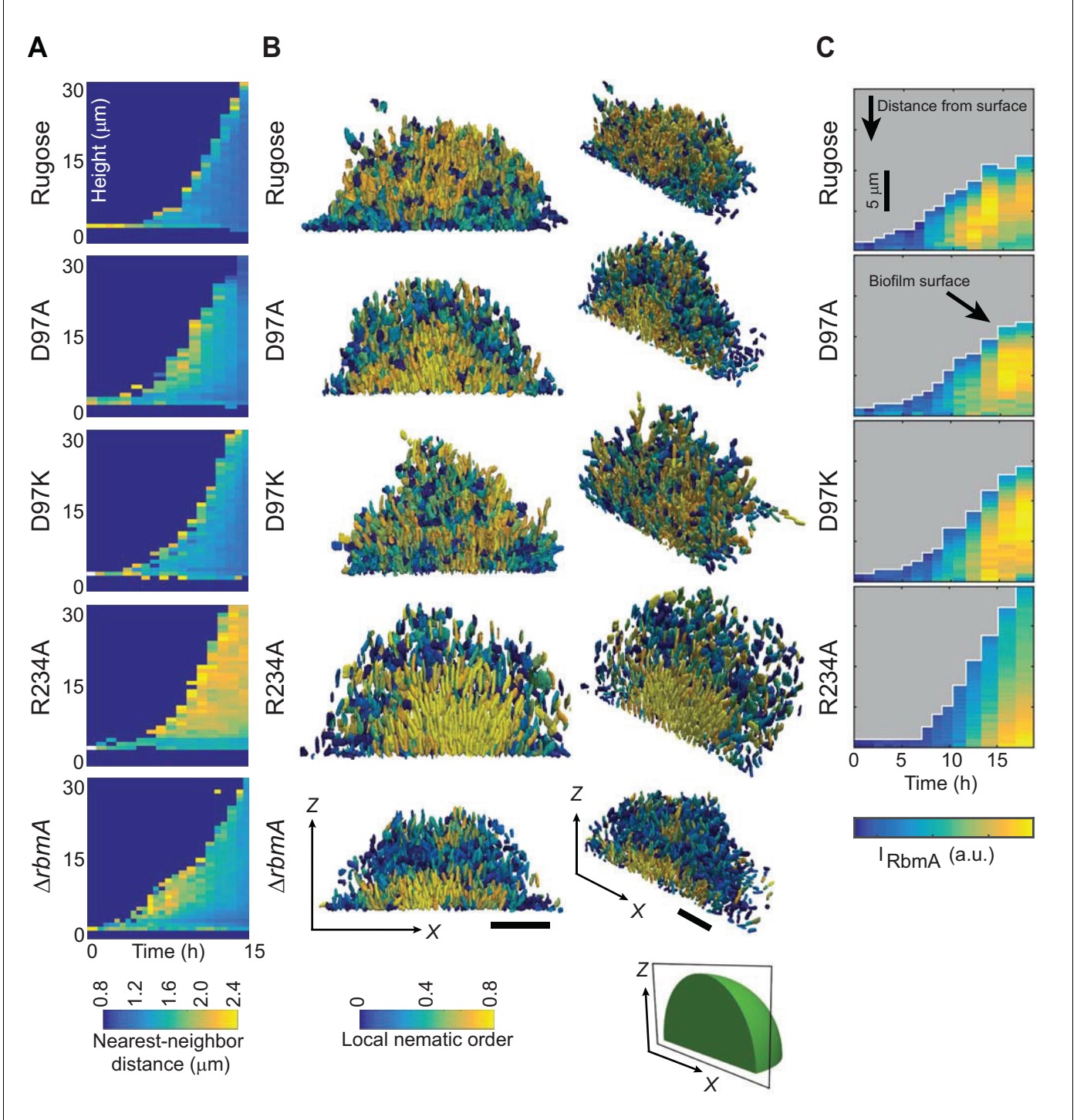

**Figure 4.** Biofilm development at single-cell resolution. (A) Average nearest-neighbor distance between individual cells, measured as a function of time and height of a biofilm colony. At each time point in this heatmap, the spatial variation of the nearest-neighbor distance inside the biofilm colony is represented as a single column, which shows the distribution of the nearest-neighbor distance between cells as a function of height in the biofilm. (B) Reconstructions of biofilms containing approximately 1500 cells, looking at the x-z-plane that cuts through the center of the biofilm colony. Each cell is color-coded according to the local nematic order in its vicinity. Scale bars, 10 μm. (C) RbmA localization within biofilms of rugose, D97A, D97K and R234A strains as a function of time. At each time point in the heatmap, the spatial variation of the intensity of the RbmA immunofluorescence ($I_{RbmA}$) is represented as a single column, which represents the RbmA abundance as a function of distance from the outer surface of the biofilm.

DOI: https://doi.org/10.7554/eLife.26163.012

*Figure 4 continued on next page*

*Figure 4 continued*

The following figure supplement is available for figure 4:

**Figure supplement 1.** Nearest-neighbor distance, vertical orientation, radial orientation and volume of individual cell in the biofilms formed by rugose parental strain and various mutants, analyzed at single-cell resolution.

DOI: https://doi.org/10.7554/eLife.26163.013

Because we observed that the FnIII-2 domain interacts directly with VPS, we analyzed the contribution of individual FnIII domains and RbmA* in colony biofilm formation. To this end, we expressed the coding regions of wild-type RbmA, isolated FnIII domains, or RbmA* in trans and examined ability of these proteins to complement a defect in colony biofilm phenotypes of the ΔrbmA strain. ΔrbmA strains harboring various expression plasmids phenocopied the ΔrbmA strain carrying vector when grown in the absence of inducer (*Figure 6—figure supplement 3*). Expression of the FnIII-2 domain in any form (prbmA, prbmA*, pFnIII-2) partially complemented the colony biofilm phenotype, indicating that FnIII-2 domain alone can contribute to biofilm architecture. Expression of FnIII-1 domain alone in the ΔrbmA strain increased colony compactness when compared to ΔrbmA harboring the vector, albeit to smaller extent, suggesting that the FnIII-1 domain alone, at least in part, may also contribute to biofilm development.

## Discussion

The importance of matrix proteins for assembly and structural integrity of biofilms is well documented in several microbial species. In *Pseudomonas aeruginosa*, CdrA adhesin directly binds to the Psl polysaccharide and facilitates cell-cell aggregation and biofilm maturation (*Borlee et al., 2010*). In *Bacillus subtilis*, amyloid-like fiber-producing protein TasA, bacterial hydrophobin BslA, and an exopolysaccharide are required for formation of mature biofilm architecture and development of a hydrophobic layer on the surface of biofilms. It is proposed that TasA, BslA, and exopolysaccharide interact to facilitate matrix assembly (*Branda et al., 2006*; *Ostrowski et al., 2011*; *Kobayashi and Iwano, 2012*; *Hobley et al., 2013*). In *V. cholerae*, biofilm formation relies on polysaccharides (VPS) and the matrix proteins RbmA, RbmC, and Bap1. The molecular interactions between matrix components and the mechanisms by which bacteria achieve spatial segregation of exopolysaccharides and proteins within the biofilm are poorly understood. This study demonstrates that a patch on the exposed outer surface of the FnIII-2 domain of RbmA binds to VPS and that RbmA has a bistable switch that influences dimerization and the formation of higher-order oligomers of RbmA with VPS. Regulation of RbmA structure by structural dynamics plays a critical role in its ability to modulate biofilm architecture (*Figure 7*).

We speculate that during initial stages of biofilm formation, when concentrations of secreted VPS are low, full-length RbmA is best able to capture cell surface-localized VPS with high avidity due to the presence of two FnIII-2 domains in the dimer. Since the binary switch in RbmA is inherently dynamic, the O-loop and D-loop states that are present at FnIII-1:FnIII-2 dimer interfaces therefore have the potential to regulate the formation of distinct higher-order RbmA-VPS structures. We observed that locking both of the FnIII-1: FnIII-2 interfaces into the closed, O-loop state with the R234A mutation enhanced the formation of VPS-dependent higher-order structures, leading to the formation of biofilm with reduced corrugation, increased cell-cell distance, and altered RbmA localization. Additionally, VPS staining in R234A strain is reduced, which could be a result from decreased accessibility of lectin targets due to strong interaction of RbmA-R234A to VPS. By contrast, the D97A/K mutations that open the protein to a monomeric state more closely resemble the native cellular

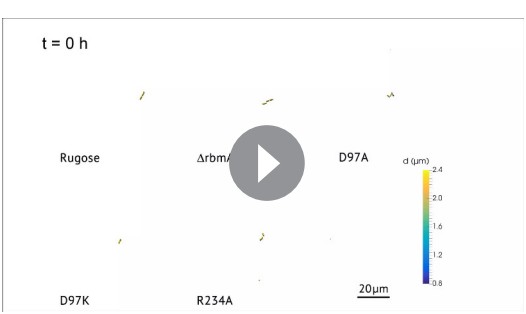

**Video 2.** Reconstructions of 3D biofilm containing approximately 1500 cells of rugose strain and various *rbmA* mutants. Color code depicts average distance to the nearest neighbor.

DOI: https://doi.org/10.7554/eLife.26163.014

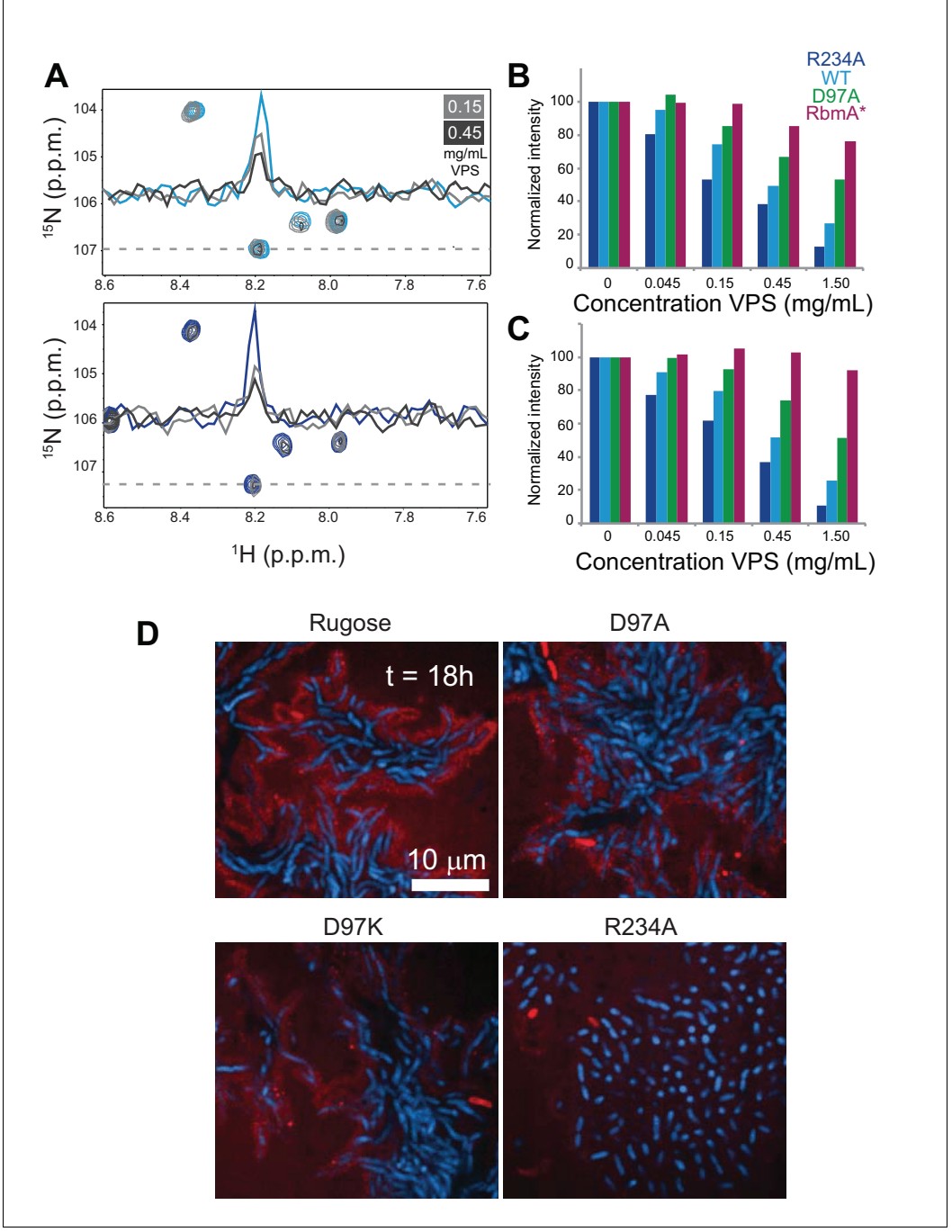

**Figure 5.** VPS binding induces formation of higher order structures in full-length RbmA dimers. (**A**) Selected regions of overlaid $^1$H-$^{15}$N TROSY-HSQC spectra of full-length $^{15}$N RbmA WT (top) and R234A (bottom). Dashed line represents horizontal trace for peak intensity analysis shown above. Blue, spectra of protein alone; Grays, addition of VPS as indicated. (**B**) Normalized peak intensities for amide and (**C**) side chain regions of $^1$H spectra of full-length $^{15}$N RbmA proteins in the presence of increasing concentrations of VPS. (**D**) VPS localization within biofilms of rugose, D97A, D97K and R234A strains at 18 hr post-inoculation. Cells were pseudo-colored blue, while VPS was pseudo-colored red.

DOI: https://doi.org/10.7554/eLife.26163.015

The following figure supplement is available for figure 5:

**Figure supplement 1.** VPS preferentially induces peak broadening in closed, dimeric forms of RbmA.

DOI: https://doi.org/10.7554/eLife.26163.016

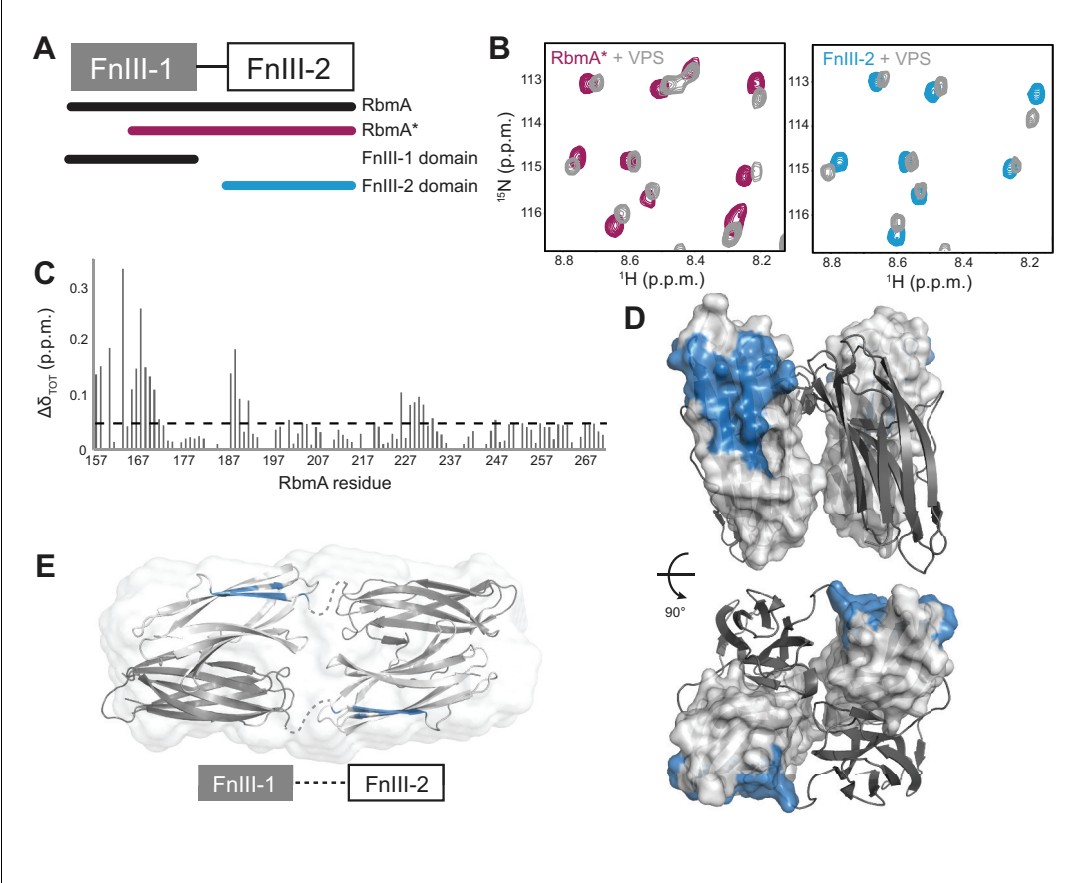

**Figure 6.** RbmA FnIII-2 domain binds VPS. (**A**) Schematic representation of RbmA structure. (**B**) Zoomed-in regions of $^1$H-$^{15}$N HSQC spectra of $^{15}$N RbmA proteins in the absence (RbmA*, magenta; FnIII-2 domain, blue) and presence (gray) of 4.5 mg/mL VPS. (**C**) Total backbone chemical shift perturbations ($\Delta\delta_{TOT}$) of $^{15}$N RbmA FnIII-2 domain upon addition of 4.5 mg/mL VPS. Dashed line, $\Delta\delta_{TOT}$ significance cutoff of 0.05 p.p.m. (**D**) FnIII-2 domain residues with significantly perturbed chemical shifts (blue) mapped onto RbmA dimer crystal structure (PDB: 4KKP). FnIII-1 domain shown in cartoon representation (gray), and the FnIII-2 domain shown in surface representation (white). (**E**) VPS-binding site mapped in blue on dimeric RbmA in its low-resolution solution envelope (*Giglio et al., 2013*) (same domain coloring as above).

DOI: https://doi.org/10.7554/eLife.26163.017

The following figure supplements are available for figure 6:

**Figure supplement 1.** Domain mapping and characterization of RbmA FnIII domains.
DOI: https://doi.org/10.7554/eLife.26163.018
**Figure supplement 2.** NMR analysis of the FnIII-2 domain R234A mutation.
DOI: https://doi.org/10.7554/eLife.26163.019
**Figure supplement 3.** Complementation by FnIII domains and RbmA*.
DOI: https://doi.org/10.7554/eLife.26163.020

organization in the biofilm. Therefore, it appears that the ability of RbmA to take on an open, D-loop state is apparently critical for governing biofilm architecture.

Enhanced interaction of RbmA with VPS could affect a network of interactions between VPS and other matrix proteins. Our earlier work showed that sustained incorporation of VPS throughout biofilms requires RbmC and that in the absence of RbmA, distribution of Bap1 and RbmC around the cluster envelope is altered (*Berk et al., 2012*). It is likely that localization of RbmC and VPS is altered in R234A strain, leading to altered biofilm architecture and stability. RbmA has also been shown to bind to glycans found in *V. cholerae* lipopolysaccharide (*Maestre-Reyna et al., 2013*). Thus, it is also possible that interaction of RbmA with other cell surface polysaccharides or biofilm matrix proteins may be governed by conformational dynamics to regulate cell-spacing and biofilm architecture.

As biofilm formation progresses, RbmA undergoes a regulated proteolytic event that shifts the entire population of RbmA to the monomeric, open D-loop state that we observed with the RbmA*

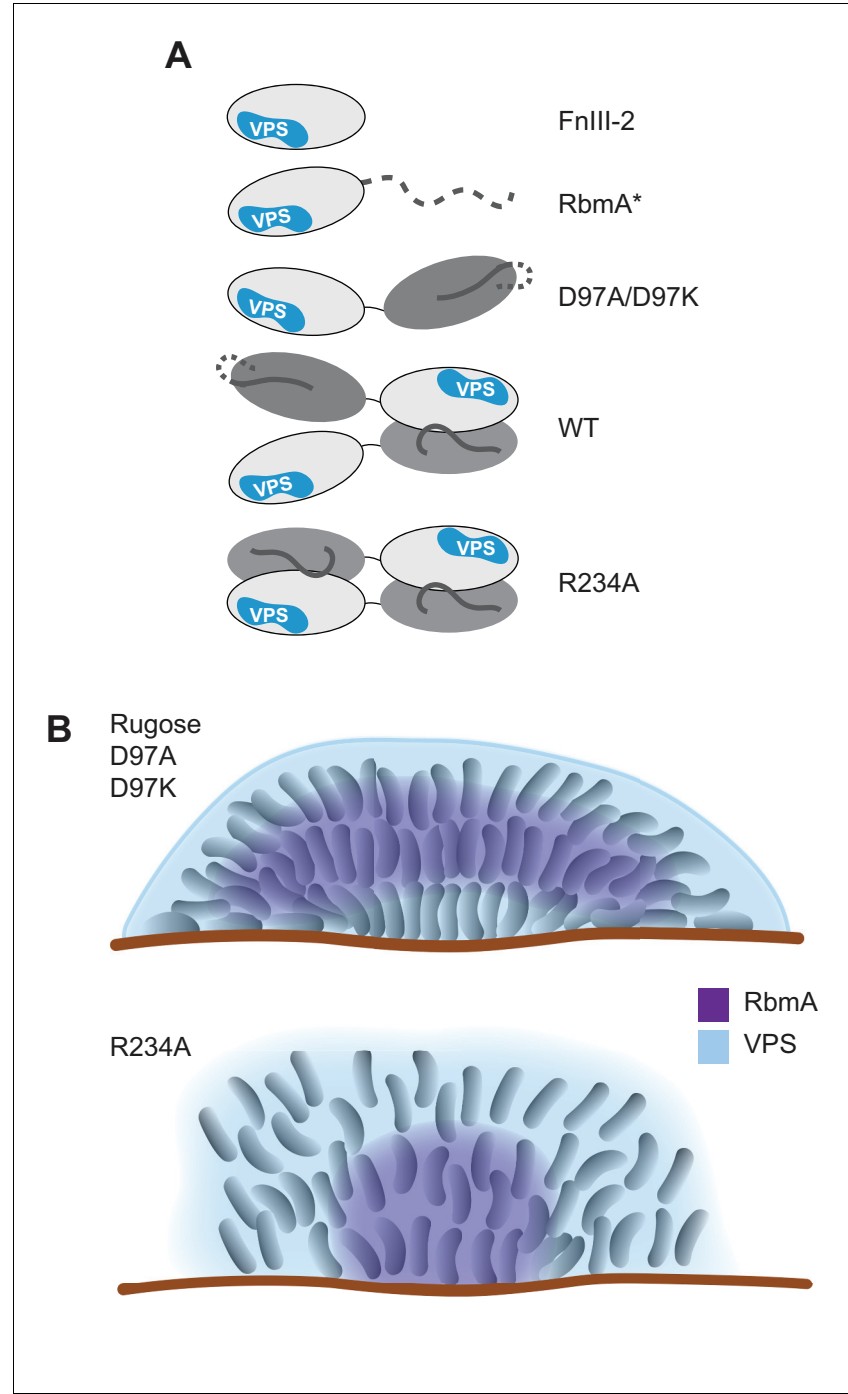

**Figure 7.** Models of various RbmA variants and their involvement in biofilm formation. (**A**) Cartoon models representing the various RbmA oligomerization and switch states. The VPS binding is indicated in blue on the FnIII-2 domain (white); dashed line indicates flexible or unfolded protein. The binary switch is represented exclusively in the open state (D-loop conformation) in the D97A/D97K mutants, and at partial occupancy in the WT protein, while the R234A mutant depicts the binary switch in its closed state (O-loop conformation). Dimeric forms of RbmA (WT, R234A) are drawn in their elongated form, as supported by small angle x-ray scattering studies in solution (*Giglio et al., 2013*). (**B**) Cell organization and RbmA localization in the biofilms of rugose, ∆*rbmA* and various *rbmA* mutants. RbmA, represented in purple, is distributed between the surface and the center of the biofilms in rugose and D97 mutants; while in R234A mutant, RbmA localizes at the center of the biofilm with decrease abundance towards the surface. VPS represented in blue and enhanced VPS localization around the biofilm cluster is represented as a thick blue line.

*Figure 7 continued on next page*

*Figure 7 continued*

DOI: https://doi.org/10.7554/eLife.26163.021

variant. We determined that RbmA FnIII-2 domain is sufficient for VPS binding in vitro and that expression of FnIII-2 domain or RbmA* alone is able to confer colony corrugation, suggesting that the FnIII-2 domain and RbmA* can cross link VPS polymers. RbmA* might be expected to exhibit reduced affinity for VPS, due to the presence of only one FnIII-2 domain in the monomer. However, at this stage of biofilm formation, increased production and accumulation of VPS in the biofilm could possibly compensate for this reduced affinity. Interestingly, it has been reported that while full-length RbmA only binds VPS-producing cells, RbmA* binds both VPS-producing and non-producing cells suggesting a mechanism by which proteolytic regulation of RbmA may help recruit cells that have not yet initiated VPS production (*Smith et al., 2015*). While we know that RbmA proteolysis occurs when cells are entering early stationary growth phase due to production of HapA, PrtV and IvaP proteases (*Smith et al., 2015*; *Hatzios et al., 2016*), where RbmA proteolysis is taking place within the biofilm during its formation and the complete consequences of proteolytic processing of RbmA remain to be determined.

Our study revealed how behavior of an individual biofilm matrix component contributes to biofilm architecture. Shifting the structural dynamics of RbmA appears to be an important mechanism regulating biofilm formation and shaping biofilm architecture. A patch on the exposed outer surface of the FnIII-2 domain constitutes the VPS binding site and binding of VPS to the outer surface of the RbmA dimer is a likely key step in mediating cell-cell adhesion in *V. cholerae* biofilms. Future studies will elucidate which components of VPS are important for interactions with RbmA, as well as defining the VPS binding site on the protein at high resolution to provide a complete picture of how RbmA interacts with VPS and contribute to the construction of biofilm matrix.

## Materials and methods

### Bacterial strains, plasmids, and culture conditions

The bacterial strains and plasmids used in this study are listed in *Table 2*. The *V. cholerae* rugose variant was used as a parent strain (*Yildiz and Schoolnik, 1999*). Mutants were generated in the rugose genetic background. *V. cholerae* and *Escherichia coli* strains were grown aerobically, at 30°C and 37°C, respectively, unless otherwise noted. Cultures were grown in Luria-Bertani (LB) broth (10 g/L tryptone, 5 g/L yeast extract, 10 g/L NaCl), pH 7.5, unless otherwise noted. LB agar medium contains 1.5% (w/v) granulated agar (BD Biosciences, Franklin Lakes, NJ). Concentrations of antibiotics and inducer used, when appropriate, were as follows: ampicillin (Ap), 100 μg/mL; rifampicin (Rif), 100 μg/mL; gentamicin (Gm), 50 μg/mL and isopropyl β-D-1-thiogalactopyranoside (IPTG), 0.1 to 0.5 mM. In-frame deletion, point mutation and GFP-tagged strains were generated according to protocols previously published (*Fong et al., 2006*; *Fong et al., 2010*; *Giglio et al., 2013*).

### Recombinant DNA techniques

DNA manipulations were carried out by standard molecular techniques according to manufacturer's instructions. Gibson Assembly master mix, restriction and DNA modification enzymes were purchased from New England Biolabs (NEB, Ipswich, MA). Polymerase chain reactions (PCR) were carried out using primers purchased from Integrated DNA Technologies (IDT, San Diego, CA) and the Phusion High-Fidelity PCR kit (NEB, Ipswich, MA), unless otherwise noted. Primers used in the present study are listed in *Supplementary file 1*. Constructs were verified by DNA sequencing (UC Berkeley DNA Sequencing Facility, Berkeley, CA). The coding regions of wild-type *rbmA*, D97A, D97K, and R234A (residues E31 to K271), *rbmA** (residues K75 to K271), FnIII-1 (residues E31 to D154) and FnIII-2 (residues E160 to K271) were cloned into the expression vectors (pGEX-6P-2 or pHisGST) by Gibson Assembly or conventional restriction cloning. For complementation, *rbmA** (residues K75 to K271), FnIII-1 (residues E31 to D154) and FnIII-2 (residues E160 to K271) were cloned into pMMB67EH, with the addition of a start codon (ATG) and the *V. cholerae* RTX type I secretion signal at the 5' and 3' ends, respectively. Chromosomal Myc tagging of *rbmA* were carried out by allelic exchange according to previous published protocol (*Berk et al., 2012*).

**Table 2.** Bacterial strains and plasmids used in this study.

| Strain or plasmid | Relevant genotype | Source |
|---|---|---|
| *E. coli* strains | | |
| CC118λ*pir* | Δ(*ara-leu*) *araD* Δ*lacX74 galE galK phoA20 thi-1 rpsE rpoB argE*(Am) *recA1* λ*pir* | (*Herrero et al., 1990*) |
| S17-1λ*pir* | Tp$^r$ Sm$^r$ *recA thi pro* r$_K^-$ m$_K^+$ RP4::2-Tc::MuKm Tn7λ*pir* | (*de Lorenzo et al., 1994*) |
| TOP10 | F− *mcrA* Δ(*mrr-hsdRMS-mcrBC*) φ80*lacZ*ΔM15 Δ*lacX74 recA1 araD139* Δ(*ara-leu*)7697 *galU galK rpsL* (Strep$^r$) *endA1 nupG* | Invitrogen |
| BL21(DE3) | F- *ompT hsdSB* (rB$^-$mB$^-$) *gal dcm* (DE3) | Invitrogen |
| *V. cholerae* strains | | |
| FY_VC_1 | Smooth wild-type, *V. cholerae* O1 El Tor A1552, Rif$^r$ | (*Yildiz and Schoolnik, 1999*) |
| FY_VC_2 | Rugose variant, *V. cholerae* O1 El Tor A1552, Rif$^r$ | (*Yildiz and Schoolnik, 1999*) |
| FY_VC_6431 | Rugose variant *rbmA*-Myc, Rif$^r$ | (*Berk et al., 2012*) |
| FY_VC_4327 | Δ*vps*-IΔ*vps*-II, deletion of *vpsA-K* and *vpsL-Q* in rugose variant, Rif$^r$ | (*Fong et al., 2010*) |
| FY_VC_105 | Δ*rbmA*, in-frame chromosomal deletion in rugose variant, Rif$^r$ | (*Fong et al., 2006*) |
| FY_VC_10035 | *rbmA*-D97A, chromosomal point mutant in rugose variant, Rif$^r$ | This study |
| FY_VC_11995 | *rbmA*-D97A-Myc, chromosomal point mutant in rugose variant, Rif$^r$ | This study |
| FY_VC_10039 | *rbmA*-D97K, chromosomal point mutant in rugose variant, Rif$^r$ | This study |
| FY_VC_11998 | *rbmA*-D97K-Myc, chromosomal point mutant in rugose variant, Rif$^r$ | This study |
| FY_VC_8795 | *rbmA*-R234A-Myc, chromosomal point mutant in rugose variant, Rif$^r$ | (*Giglio et al., 2013*) |
| FY_VC_10283 | Δ*hapA*Δ*prtV*Δ*ivaP*, in-frame chromosomal deletion in rugose variant, Rif$^r$ | This study |
| FY_VC_240 | Rugose-*gfp*, *V. cholerae* O1 El Tor A1552, rugose variant, Rif$^r$ Gm$^r$ | (*Beyhan and Yildiz, 2007*) |
| FY_VC_224 | Δ*rbmA-gfp*, Rif$^r$ Gm$^r$ | (*Fong et al., 2006*) |
| FY_VC_10084 | *rbmA*-D97A-*gfp*, Rif$^r$ Gm$^r$ | This study |
| FY_VC_10086 | *rbmA*-D97K-*gfp*, Rif$^r$ Gm$^r$ | This study |
| FY_VC_8832 | *rbmA*-R234A-*gfp*, Rif$^r$ Gm$^r$ | This study |
| Plasmids | | |
| pGP704*sacB*28 | pGP704 derivative, *mob/oriT sacB*, Ap$^r$ | G. Schoolnik |
| pFY-4183 | pGP-*rbmA*-D97A, for chromosomal point mutation, Ap$^r$ | This study |
| pFY-4523 | pGP-*rbmA*-D97A-Myc, for chromosomal point mutation, Ap$^r$ | This study |
| pFY-4185 | pGP-*rbmA*-D97K, for chromosomal point mutation, Ap$^r$ | This study |
| pFY-4524 | pGP-*rbmA*-D97K-Myc, for chromosomal point mutation, Ap$^r$ | This study |
| pFY-3509 | pGP-Δ*hapA*, Ap$^r$ | This study |
| pFY-3512 | pGP-Δ*prtV*, Ap$^r$ | This study |
| pFY-4277 | pGP-Δ*ivaP*, Ap$^r$ | This study |
| pMMB67EH | Low copy number IPTG inducible vector, Ap$^r$ | (*Fürste et al., 1986*) |
| pFY-4260 | p*rbmA*, pMMB67EH containing wild-type *rbmA* in-frame with *rtx* type I secretion signal, no native type II secretion signal, Ap$^r$ | This study |
| pFY-4261 | pFnIII-1, pMMB67EH containing FnIII-1 in-frame with *rtx* type I secretion signal, Ap$^r$ | This study |
| pFY-4262 | pFnIII-2, pMMB67EH containing FnIII-2 in-frame with *rtx* type I secretion signal, Ap$^r$ | This study |
| pFY-4263 | p*rbmA**, pMMB67EH containing *rbmA** in-frame with *rtx* type I secretion signal, Ap$^r$ | This study |
| pGEX-6P-2 | IPTG-inducible vector for expression of recombinant proteins with N-terminal GST tag, cleavable by PreScission protease, Ap$^r$ | GE |
| pHisGST | IPTG-inducible vector for expression of recombinant proteins with N-terminal His$_6$-GST tags, cleavable by TEV protease, Ap$^r$, | (*Xu et al., 2015*) |
| pFY-1429 | pGEX-*rbmA*, no native type II secretion signal Ap$^r$ | This study |

*Table 2 continued on next page*

*Table 2 continued*

| Strain or plasmid | Relevant genotype | Source |
|---|---|---|
| pFY-4194 | pHisGST-noSP-*rbmA*-D97A, no native type II secretion signal, Ap[r] | This study |
| pFY-4196 | pHisGST-noSP-*rbmA*-D97K, no native type II secretion signal, Ap[r] | This study |
| pFY-3491 | pGEX-noSP-*rbmA*-R234A, no native type II secretion signal, Ap[r] | This study |
| pFY-4219 | pHisGST-noSP-*rbmA** no native type II secretion signal, Ap[r] | This study |
| pFY-3551 | pHisGST-FnIII-1, Apr | This study |
| pFY-3066 | pHisGST-FnIII-2, Ap[r] | This study |
| pFY-3435 | pHisGST-FnIII-2-R234A, Ap[r] | This study |
| pUX-BF13 | *ori*R6K helper plasmid, *mob/oriT*, provides Tn7 transposition function in trans, Ap[r] | (*Bao et al., 1991*) |
| pMCM11 | pGP704::mTn7-*gfp*, Gm[r] Ap[r] | M. Miller and G. Schoolnik |
| pNUT542 | P$_{tac}$_*sfgfp* expression plasmid, Gm[r] | This study |
| pNUT1029 | P$_{tac}$_*mRuby* expression plasmid, Gm[r] | This study |

DOI: https://doi.org/10.7554/eLife.26163.022

## VPS and protein purification

Purification of VPS from *V. cholerae* rugose strain was carried out using a previously published protocol (*Yildiz et al., 2014*). For protein purification, recombinant *E. coli* BL21 (DE3) cells carrying expression plasmids were grown at 37°C in M9 minimal medium (6 g/L $Na_2HPO_4$, 3 g/L $KH_2PO_4$, 0.5 g/L NaCl, 0.1 mM $CaCl_2$, 1 mM $MgSO_4$, 3 g/L glucose, 1 g/L $^{15}NH_4Cl$) supplemented with MEM vitamin solution (Sigma, St. Louis, MO) until optical density at 600 nm ($OD_{600}$) reached 0.6 to 0.8. Induction was carry out by adding IPTG to a final concentration of 0.5 mM and the cultures were grown overnight (16 to 18 hr) at 18°C. To produce uniformly labeled $^{13}C$, $^{15}N$ FnIII-2 domain for chemical shift assignments, 3 g/L of $^{13}C$-glucose was used in place of glucose in the M9 medium. Cells were harvested by centrifugation and lysed in buffer containing 50 mM Tris (pH 8.0), 1 M NaCl, 0.5% (v/v) Tween-20 and Complete Protease Inhibitor (Roche, Indianapolis, IN) via sonication. Proteins were purified with Fast Protein Liquid Chromatography (FPLC) using a GSTPrep FF 16/10 column containing glutathione sepharose (GE Healthcare Bio-Sciences, Marlborough, MA) and a BioLogic DuoFlow FPLC system (Bio-Rad, Hercules, CA). The column was first washed with buffer containing 50 mM Tris (pH 8.0), 0.25 M NaCl, 0.05% (v/v) Tween-20 and 0.5 mM DTT, followed by another wash with buffer containing 50 mM Tris (pH 8.0), 0.25 M NaCl, and 0.5 mM DTT. Elution was carried out with buffer containing 50 mM Tris (pH 8.0), 0.25 M NaCl, and 3 g/L glutathione. Buffer exchange was carried out with the eluted protein and the digestion buffer using Amicon Ultra-15 centrifugal filter units (Millipore) and subsequent proteolysis of the protein tag was carried out overnight at 4°C with either PreScission protease (for recombinant proteins expressed from pGEX-6P-2) or tobacco etch virus (TEV) protease (for recombinant proteins expressed from pHisGST). The digestion buffer for PreScission protease contains 50 mM Tris (pH 7.5) and 0.15 M NaCl, while the digestion buffer for TEV protease contains 50 mM Tris (pH 8.0), 0.5 mM EDTA (pH 8.0) and 1 mM DTT. After proteolysis, cleaved tag and the protease were remove by another round of FPLC using either the GSTPrep FF 16/10 column (for recombinant proteins expressed from pGEX-6P-2) or HisPrep FF 16/10 containing nickel sepharose (for recombinant proteins expressed from pHisGST). Untagged protein was concentrated using Amicon Ultra-15 centrifugal filter units and further purified using size-exclusion chromatography on Superdex 75 16/600 (GE Healthcare Bio-Sciences, Marlborough, MA) in NMR buffer (10 mM HEPES, pH 7.0, and 100 mM NaCl).

## Analysis of biofilm formation

Colony biofilm analysis were carried out as described previously (*Fong et al., 2010*). Images of colonies were acquired after 2–5 days of growth at 25°C using a Zeiss Stemi 2000-C microscope equipped with Zeiss AxioCam ERc 5 s Microscope Camera. When needed ampicillin (100 μg/mL) and IPTG (100 μM) were added. Experiments were repeated with 2 biological replicates.

For time-lapse and thin section microscopy studies, two LB agar layers (pH 7.5; with 20 μg/mL Congo red) were poured to depths of 4.5 mm (bottom) and 1.5 mm (top). Overnight cultures were diluted 1:200 in fresh LB, then 3 μL were spotted onto the top agar layer and the bacteria were allowed to grow for 2 to 5 days (dark; 25°C; >95% humidity). For thin sectioning, the bacterial colonies were imaged using a Keyence VHX 1000 digital microscope before they were covered by an additional 1.5 mm agar layer. Colonies sandwiched between two 1.5 mm agar layers were lifted from the bottom layer and soaked for 4 hr in 50 mM L-lysine in phosphate buffered saline (PBS) (pH 7.4) at 4°C, then fixed in 4% paraformaldehyde, 50 mM L-lysine, PBS (pH 7.4) for 4 hr at 4°C, then overnight at 37°C. Fixed samples were washed twice in PBS and dehydrated through a series of ethanol washes (25%, 50%, 70%, 95%, 3 × 100%) for 60 min each. Samples were cleared via three 60 min incubations in Histoclear-II (National Diagnostics), and then infiltrated for 4 hr at 55°C with 100% paraffin wax (Paraplast Xtra), which polymerized overnight at 4°C. Trimmed blocks were sectioned (10 μm-thick sections perpendicular to the plane of the bacterial colony spots), floated onto water at 45°C, and collected onto slides. Slides were air-dried overnight, heat-fixed on a hotplate for 1 hr at 52°C, and rehydrated in the reverse order of processing. Rehydrated samples were immediately mounted in TRIS-buffered DAPI:Fluorogel (Electron Microscopy Sciences). Time-lapse and thin section microscopy studies were carried out with three biological replicates.

For flow-cell biofilm studies, Ibidi μ-Slide VI0.4 (Ibidi 80601, Ibidi LLC, Verona, WI) flow-cell chambers were inoculated with 100 μL of overnight-grown cultures, normalized to an $OD_{600}$ of 0.02. Flow-cell chambers were incubated at room temperature for 1 hr, then flow of diluted LB (0.2 g/L tryptone, 0.1 g/L yeast extract, 10 g/L NaCl, pH 7.5) was initiated at a rate of 20 mL/h and continued for up to 24 hr. Confocal images were obtained on a Zeiss LSM 5 Pascal laser scanning confocal microscope (Zeiss, Dublin, CA). Images were obtained with a 40 dry objective and were processed using Imaris software (Biplane, South Windsor, CT). Confocal laser scanning microscopy (CLSM) were carried out with at least two biological replicates. Colony forming units (CFU) from the effluent of flow-cells in the CLSM experiments were quantified by dilution plating and were repeated with two biological replicates.

For single-cell resolution microscopy of biofilms, GFP-expression plasmid pNUT542 was introduced into the relevant strains, which conferred a gentamycin resistance. The strains were grown in M9 minimal medium, supplemented with 0.5% (w/v) glucose and 30 μg/mL gentamycin, to mid-exponential growth phase, before introducing into microfluidic channels, which consisted of a trench in a polydimethylsiloxane block that was $O_2$-plasma-bonded to a glass coverslip. The resulting flow chambers had a width of 500 μm, height of 100 μm, and length of 7 mm. After the cultures were introduced into the channels, the channels were incubated at 24°C for 1 hr without any flow, to allow cells to attach to the bottom glass surface of the channels. The flow was then set to 1 μL/min, corresponding to an average flow speed of 330 μm/s in the channel, for the remainder of the experiment. While the medium was flown through the channel, the biofilm architecture of the different strains was imaged every 30–60 min for approximately 20 hr. Images were acquired with an Olympus 100x objective with numerical aperture of 1.35, using a Yokogawa spinning disk confocal scanner and laser excitation at 488 nm. Images were acquired at spatial resolution of 63 nm in the *xy* plane and 400 nm along the *z* direction.

To detect all single cells and measure architectural properties of the biofilms grown in flow chambers, images were analyzed using custom code developed in Matlab, which was significantly improved from previous single-cell imaging methods (*Drescher et al., 2016*). The custom code is available online at GitHub (*Hartmann, 2017*; a copy is archived at https://github.com/elifesciences-publications/single-cell-analysis). Briefly, cells were automatically identified using edge detection algorithms, before clumps were broken up using a watershed algorithm. Cellular orientations were obtained by mapping an ellipsoid onto each cell using principal component analysis. Nematic refers to liquid crystal 'state' that some materials can exhibit under certain conditions and such materials assume thread-like formations. Molecules in this state do not exhibit any positional order but they exhibit a certain degree of orientational order. The vectors of the major cell axes were used to determine the value of the local nematic order parameter, defined as, $S = <3/2 \, (\mathbf{n}_i \cdot \mathbf{n}_j)^2 - 1/2>$, where $\mathbf{n}_i$ is the orientation vector of a particular cell and $\mathbf{n}_j$ are the orientation vectors of cells in the local vicinity, within a local vicinity defined by a sphere of radius 3 μm around each cell. Vertical alignment of cells was calculated as the angle of each cell with the vertical *z*-axis. The average distance to the

nearest neighbor were calculated for each cell based on cell-centroid coordinates. Single-cell resolution microscopy was performed on biofilm colonies with three biological replicates per strain.

For localization of RbmA and VPS during biofilm growth, cells expressed the red fluorescent protein mRuby from the plasmid pNUT1029, driven by a constitutively active $P_{tac}$ promoter. For RbmA localization, biofilms were grown in M9 medium containing 1 μg/mL of c-Myc tag monoclonal antibody (9E10) conjugated to Alexa Flour 488 (Thermo Scientific) and 1 mg/mL of filter-sterilized bovine serum albumin (BSA). For VPS localization, biofilms were grown in M9 medium containing 20 μg/mL wheat-germ agglutinin (WGA) conjugated to FITC and 20 μg/mL Concanavalin A conjugated to FITC (Sigma), and 1 mg/mL of filter-sterilized BSA.

## Nuclear magnetic resonance (NMR) spectroscopy

All NMR experiments were conducted at 35°C on either a Varian 600 MHz spectrometer equipped with a 5 mm $^1$H, $^{13}$C, $^{15}$N cryoprobe or a Bruker 900 MHz spectrometer equipped with a 5 mm CP TCI ($^1$H, $^{13}$C, $^{15}$N, $^2$H) cryoprobe. All NMR data were processed using NMRPipe/NMRDraw (*Delaglio et al., 1995*). Chemical shift assignments were made with NMRViewJ RunAbout (*Johnson, 2004*) using data obtained from the following 3D triple resonance experiments acquired on 300 μM $^{13}$C/$^{15}$N labeled FnIII-2 domain: HNCO, HNCACB, CBCA(CO)NH, and C(CO)NH-TOCSY spectra. 93% of the total non-proline residues were assigned, including 99% of the non-proline residues originating from the FnIII-2 domain itself. Chemical shifts are available at the BMRB database (BMRB 27130). Chemical shift assignments were validated by comparing secondary structure predictions from TALOS+ (*Shen et al., 2009*) to existing crystal structures (*Giglio et al., 2013*). Chemical shift assignments for $^{15}$N R234A FnIII-2 domain were made by re-assignment from WT FnIII-2 domain using the Minimal Chemical Shift Analysis function in NMRViewJ. Purified, lyophilized VPS (*Yildiz et al., 2014*) was resuspended to 50 mg/mL in NMR buffer (10 mM HEPES, pH 7.0, and 100 mM NaCl) with 3–5 short pulses of a benchtop vortexer to promote solubilization. VPS titration experiments were performed by collecting $^1$H-$^{15}$N HSQC spectra at 35°C on either a 600 MHz with 100 μM $^{15}$N FnIII-2 domain (WT or R234A) or $^1$H-$^{15}$N TROSY-HSQC or 1D $^1$H spectra at 900 MHz on 100 μM $^{15}$N full-length RbmA proteins (WT, R234A, D97A, and RbmA*) in the presence of increasing concentrations of VPS. Samples were brought up to 300 μL with NMR buffer and adjusted with D$_2$O to a final concentration of 10% (v/v). All HSQC titration data were visualized and analyzed with NMRViewJ. Chemical shift perturbations defined by the equation $\Delta\delta_{TOT} = [(\Delta\delta^1H)^2 + (\chi(\Delta\delta^{15}N)^2]^{1/2}$ and normalized with the scaling factor $\chi = 0.5$ (*Johnson, 2004*). TROSY-HSQC spectra were processed with 48 Hz shifts in both $^{15}$N and $^1$H dimensions to align HSQC and TROSY-HSQC spectra. 1D $^1$H NMR data were processed and analyzed with MestReNova. A decrease in peak intensity (i.e. broadening) can arise from several different phenomena, including formation of high molecular weight species with poor relaxation properties or chemical exchange between two or more species on an intermediate timescale; these properties and considerations for the interpretation of NMR data are explained in more detail in the reviews on NMR spectroscopy (*Kleckner and Foster, 2011*; *Kwan et al., 2011*).

## SEC-MALS

All experiments were performed on a Superdex S75 10/300 GL column (GE Healthcare) with a sample injection volume of 250 μL and a flow rate of 0.5 mL/min at 4°C. The volume of the sample loop was 100 μL. Molecular weight markers used to calibrate the column were: albumin (67 kDa), ovalbumin (43 kDa), chymotrypsinogen A (25 kDa), and ribonuclease A (13.7 kDa) from the Low Molecular Weight Gel-filtration Calibration kit (GE Healthcare). Absolute molecular weight calculations were obtained by static light scattering in line with size exclusion chromatography using the Wyatt Optilab T-rEX refractometer and miniDAWN Treos multiangle light scattering system at 4°C. Protein concentrations were monitored by the refractometer and light scattering unit directly after the gel filtration column. Absolute molecular weights were determined using ASTRA version 6.0 (Wyatt Technologies).

## Limited proteolysis and mass spectrometry

Limited digestion of purified proteins was performed at 1.5 mg/mL in 10 mM HEPES, pH 7.0, and 100 mM NaCl with sequencing-grade trypsin (Promega) for one hour at room temperature 1:150

mass (w/w) ratios. Reactions were quenched with addition of an equal volume of 2X SDS Laemmli buffer (Bio-Rad) and samples were boiled at 95°C for 5 min. Digested fragments were resolved by 18% SDS-PAGE and detected by Coomassie stain. For mass determination, proteolysis reactions were quenched in parallel by addition of formic acid to a final concentration of 1% (v/v). Samples were desalted and separated by HPLC (Surveyor; Thermo Finnegan) on a Proto 300 C4 reverse-phase column with a 100 mm x 2.1 mm inner diameter and 5 μm particle size (Higgins Analytical, Inc.) using a mobile phase consisting of solvent A (0.1% formic acid in HPLC grade water) and solvent B (0.1% formic acid in acetonitrile). Samples were then analyzed on an LTQ Orbitrap linear ion mass spectrometer system (Thermo Finnegan). Proteins were detected by full-scan MS mode (over an m/z of 300–2000) in positive mode with the electrospray voltage set to 5 kV. Mass measurements of deconvoluted electrospray ionization mass spectra of the reversed-phase peaks were generated by Magtran software.

## Western blot analysis

Total proteins (10 μg) extracted from whole-cell (WC) or precipitated from culture supernatant (CS) were separated on 10% SDS-PAGE and transferred to PVDF for Western blot analysis, following previously described protocol (*Giglio et al., 2013*). For planktonic cells, overnight cultures were diluted 1:200 and grown at 30°C in LB medium with shaking at 200 rpm for the indicated amount of time. Culture supernatant was separated from the cells by centrifugation at 5000 x *g*. Whole-cell samples were prepared by resuspending the cell pellets in 2% (w/v) SDS. The culture supernatant fractions were collected and filtered through 0.22 μm filters to remove any residual cells. Bovine serum albumin (BSA) (700 μg) was added to 20 mL of each culture supernatant fractions as an additional loading control. Total protein in the culture supernatant was precipitated with 13% (v/v) trichloroacetic acid (TCA) at 4°C overnight, followed by centrifugation at 45,000 x *g* for 1 hr. The protein pellets from the culture supernatant were washed with 2 mL ice-cold acetone and resuspended in 1x PBS. For biofilm cells, overnight cultures were diluted to OD$_{600}$ of 0.02 with diluted LB medium, inoculated into Ibidi μ-Slide and grown as described for CLSM experiments. Biofilms cells were pelleted via centrifugation after harvesting from the chambers, resuspended in 2% (w/v) SDS and 5 μg of total proteins were separated on 12% SDS-PAGE and transferred to PVDF for Western blot analysis. Protein concentrations were estimated using a Pierce BCA protein assay kit (Thermo Fisher Scientific, Waltham, MA) and BSA as standard. Polyclonal rabbit anti-RbmA serum, generated against purified RbmA protein (*Berk et al., 2012*), was used at a dilution of 1:1500 to detect RbmA. Additional loading controls where appropriate were used, BSA in the culture supernatant samples and RNA polymerase in the whole-cell samples were detected using 1:500-diluted polyclonal rabbit anti-BSA antibody (Thermo Fisher Scientific) and 1:2000-diluted monoclonal mouse anti-RNAP (BioLegend Neoclone, San Diego, CA), respectively. Anti-RNAP was also used to verify that the culture supernatant samples were not contaminated with whole-cell proteins. For samples harvested from colony biofilms, rabbit anti-OmpU polyclonal antiserum was used as a loading control. Secondary goat anti-rabbit and anti-mouse IgG conjugated to horseradish peroxidase (Santa Cruz Biotechnology, Paso Robles, CA) was used at a dilution of 1:2500. The SuperSignal West Pico chemiluminescent substrate (Thermo Fisher Scientific) and a Bio-Rad ChemiDoc MP imaging system were used for detection and capturing of the Western blot signals. Western blot analysis was carried out with two biological replicates

## Quantitative and statistical analyses

Biomass, average and maximum biofilm thicknesses were calculated from at least five biofilm images using COMSTAT software package (*Heydorn et al., 2000*). Statistical significance was determined using Student's *t* test (two-tailed, unpaired).

## Acknowledgements

We thank Benjamin Abrams from UCSC Life Sciences Microscopy Center for his technical support, as well as Hsiau-Wei Lee from the UCSC NMR Facility and Qiangli Zhang of the UCSC Mass Spectrometry Facility, which received support from the W.M. Keck Foundation (Grant 001768) and the NIH Center for Research Resources (Grant S10 RR020939). We thank Jeffrey Pelton at the Central California 900 MHz QB3-Berkeley NMR Facility for his assistance with data collection and analysis. Work on

the 900 MHz QB3 NMR spectrometer was funded by NIH grant GM68933. We thank Kassidy Hebert for help with RbmA purification and RbmA quantification experiments. This work was supported by NIH grants R01 AI103369 (L.E.P.D), R01 GM107069 (C.L.P.) and R01 AI055987 (F.H.Y.), as well as Human Frontier Science Program grant CDA00084/2015 C and the Max Planck Society (K.D.). A.K. M. was supported by NIH fellowship F31 CA189660.

## Additional information

### Funding

| Funder | Grant reference number | Author |
| --- | --- | --- |
| National Institute of Allergy and Infectious Diseases | RO1AI055987 | Fitnat H Yildiz |
| National Institute of General Medical Sciences | GM107069 | Carrie L Partch |
| Human Frontier Science Program | CDA00084/2015-C | Knut Drescher |
| National Institute of General Medical Sciences | CA189660 | Alicia K Michael |
| National Institute of Allergy and Infectious Diseases | R01 AI103369 | Lars EP Dietrich |

The funders had no role in study design, data collection and interpretation, or the decision to submit the work for publication.

### Author contributions

Jiunn CN Fong, Conceptualization, Resources, Supervision, Validation, Investigation, Visualization, Methodology, Writing—original draft, Project administration, Writing—review and editing; Andrew Rogers, Conceptualization, Validation, Investigation, Visualization, Methodology, Writing—original draft; Alicia K Michael, Nicole C Parsley, William-Cole Cornell, Yu-Cheng Lin, Praveen K Singh, Investigation, Methodology; Raimo Hartmann, Software, Investigation, Methodology; Knut Drescher, Software, Formal analysis, Supervision, Funding acquisition, Investigation, Methodology, Writing—original draft, Project administration, Writing—review and editing; Evgeny Vinogradov, Resources; Lars EP Dietrich, Resources, Formal analysis, Supervision, Funding acquisition, Methodology, Project administration; Carrie L Partch, Conceptualization, Formal analysis, Supervision, Funding acquisition, Validation, Investigation, Visualization, Methodology, Writing—original draft, Project administration; Fitnat H Yildiz, Conceptualization, Resources, Formal analysis, Supervision, Funding acquisition, Validation, Investigation, Visualization, Methodology, Writing—original draft, Project administration, Writing—review and editing

### Author ORCIDs

Jiunn CN Fong, http://orcid.org/0000-0003-3933-7885
William-Cole Cornell, http://orcid.org/0000-0002-8927-1813
Praveen K Singh, http://orcid.org/0000-0002-0254-7400
Raimo Hartmann, http://orcid.org/0000-0002-4924-6402
Lars EP Dietrich, http://orcid.org/0000-0003-2049-1137
Carrie L Partch, http://orcid.org/0000-0002-4677-2861
Fitnat H Yildiz, http://orcid.org/0000-0002-6384-7167

### Decision letter and Author response

Decision letter https://doi.org/10.7554/eLife.26163.025
Author response https://doi.org/10.7554/eLife.26163.026

## Additional files

### Supplementary files
- Supplementary file 1. List of primers used in this study for generating various strains and plasmids.
  DOI: https://doi.org/10.7554/eLife.26163.023
- Transparent reporting form
  DOI: https://doi.org/10.7554/eLife.26163.024

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
