## [Decision Letter]

Thank you for submitting your article "Structural dynamics of RbmA governs plasticity of *Vibrio cholerae* biofilms" for consideration by *eLife*. Your article has been favorably evaluated by Philip Cole (Senior Editor) and four reviewers, one of whom, Dianne K Newman (Reviewer #1), is a member of our Board of Reviewing Editors. The following individuals involved in review of your submission have agreed to reveal their identity: Alex Persat (Reviewer #2); Matthew R Parsek (Reviewer #3).

The reviewers have discussed the reviews with one another and the Reviewing Editor has drafted this decision to help you prepare a revised submission.

Consensus Review:

Your work has been reviewed by three individuals with expertise in the biofilm matrix, biofilm biomechanics, and NMR. We all agree this work represents an exciting step forward for the biofilm field, and look forward to seeing it published in *eLife*. That said, we would like you to pay attention to the following major points when revising the manuscript for publication. Though careful revision might suffice in addressing the first two comments, please consider additional limited experimentation if this would help you respond to these concerns even better:

1) Connect the molecular to the cellular scale by investigating how dimerization and VPS binding play a role in biofilm structure: explain how RbmA dimerization and binding to VPS modulates biofilm architecture at the larger scale. From Berk et al., we know that the spatial patterning of RbmA and VPS are an important part of biofilm architecture. Given the obvious phenotype of R234A in nearest neighbor distances and its increased affinity to VPS, one could expect a dramatic change in the distribution of both RbmA and VPS, although there might be other hypothesis. To understand this, one possibility would be to use the labeling system the authors described previously (Berk et al) to visualize the spatial distribution of the different matrix components in single cell level visualizations.

2) Provide more information on the NMR data collection and processing. Information on extent of signal assignment and whether the assignment was deposited in a suitable repository should be included in the manuscript. It appears that 3mm tubes were used to perform the NMR experiments but it should be explicitly stated if the probes and tube were of 3 mm. It is not clear if only the 3D experiments mentioned were used for performing the signal assignment. No indication of whether those data were deposited in an appropriate repository, for example BMRB, degree of completeness of the assignment… the only hint we have is derived indirectly from Figure 5. Is there an assignment for the region of residue 97? Having the assignment it is possible to measure the dynamics of the residues by T1, T2 and hetero-NOE and this would indeed tell us if we are observing changes in structural dynamics instead of equilibrium shift.

The authors mention that the two states observed in the X-ray data are in fast exchange in the NMR data but that statement is only marginally supported by the data reported in Figure 1. Figure 1 shows discreet positions for discreet mutants. Shift changes are collinear, which is compatible with two-state fast exchange but does not prove that there are only two states, nor that they are in any way related with what was observed crystallographically. The authors mention that RbmA samples both O and D loop states. This can be confirmed by collecting data at different temperatures, which would change the equilibrium position and the exchange kinetics.

3) Clarify more explicitly in the Introduction what the question is that this manuscript seeks to address. Explain clearly what the novelty is. Explain how your study extends the work of Giglio 2013. If this is the first study to show at the molecular level how higher-order structure assemble in the matrix, state that. If not, explain how is it builds on other efforts of this type. In the Discussion, remind the reader of what makes this work particularly significant. Comment on how generalizable the results might be.

4) Fix figures so they can be read by a color-blind person. Helpful guidelines can be found here: http://www.nature.com/nmeth/journal/v8/n6/full/nmeth.1618.html

---

## [Author Response]

*Consensus Review:*

*Your work has been reviewed by three individuals with expertise in the biofilm matrix, biofilm biomechanics, and NMR. We all agree this work represents an exciting step forward for the biofilm field, and look forward to seeing it published in eLife. That said, we would like you to pay attention to the following major points when revising the manuscript for publication. Though careful revision might suffice in addressing the first two comments, please consider additional limited experimentation if this would help you respond to these concerns even better:*

*1) Connect the molecular to the cellular scale by investigating how dimerization and VPS binding play a role in biofilm structure: explain how RbmA dimerization and binding to VPS modulates biofilm architecture at the larger scale. From Berk et al., we know that the spatial patterning of RbmA and VPS are an important part of biofilm architecture. Given the obvious phenotype of R234A in nearest neighbor distances and its increased affinity to VPS, one could expect a dramatic change in the distribution of both RbmA and VPS, although there might be other hypothesis. To understand this, one possibility would be to use the labeling system the authors described previously (Berk et al) to visualize the spatial distribution of the different matrix components in single cell level visualizations.*

We agree with the reviewers that analysis of distribution of RbmA and VPS would provide better understanding of biofilm matrix construction and function of RbmA. We have performed the requested experiments. We found striking differences between the mutants and included new results in the revised manuscript. These data are included in Figure 4 and Figure 5, and new results are included in the text.

We have also included more in-depth discussion on the roles of RbmA in dimerization and VPS binding on biofilm structure.

*2) Provide more information on the NMR data collection and processing. Information on extent of signal assignment and whether the assignment was deposited in a suitable repository should be included in the manuscript. It appears that 3mm tubes were used to perform the NMR experiments but it should be explicitly stated if the probes and tube were of 3 mm. It is not clear if only the 3D experiments mentioned were used for performing the signal assignment. No indication of whether those data were deposited in an appropriate repository, for example BMRB, degree of completeness of the assignment… the only hint we have is derived indirectly from Figure 5. Is there an assignment for the region of residue 97? Having the assignment it is possible to measure the dynamics of the residues by T1, T2 and hetero-NOE and this would indeed tell us if we are observing changes in structural dynamics instead of equilibrium shift.*

The chemical shift assignments for the isolated FnIII-2 domain of RbmA were deposited in the BMRB database (BMRB 27130) to be released upon publication. We assigned 93% of the total non-proline backbone amides in the protein construct (including 99% of the non-proline residues originating from the FnIII-2 domain itself). We have added more information about data collection (including the 5 mm diameter tube used in our experiments) and clarified the exact spectra used for chemical shift assignment in the Materials and methods section of the manuscript.

As stated in the manuscript, we were only able to obtain assignments for the isolated FnIII-2 domain. This was largely due to issues with aggregation of the isolated FnIII-1 domain (Figure 6—figure supplement 1, black spectrum) and some challenges working with the full-length proteins that limited us to working with relatively low concentrations (100 µM) in order to prevent formation of soluble aggregates that limited signal to noise. Our spectra of the full-length proteins (WT, R234A and D97A/D97K) were all dominated by peaks that appear to originate from the FnIII-2 domain. Moreover, overlays of the isolated FnIII-2 domain (blue) with the spontaneously cleaved version of the full-length protein (orange) or engineered, truncated version RbmA* (magenta) in Figure 6—figure supplement 1 also showed that the FnIII-2 domain represents the functionally relevant domain in these cleaved species. Therefore, while we do not have assignments for the region around residue 97 in the FnIII-1 domain loop, we believe that our NMR studies that focus on residues in the FnIII-2 domain faithfully report on changes in the intact RbmA protein. We agree that having complete chemical shift assignments would open up new avenues for experimentation, but we are currently limited by technical challenges with the full-length protein.

Therefore, without assignments for the FnIII-1 domain loop, we cannot easily address protein dynamics in the full-length protein using T1, T2 and hetero-NOE experiments by NMR spectroscopy. Although this is an interesting idea, it’s not clear that there would be enough change in the flexibility of individual domains between the open and closed states given the very short length of the interdomain linker (only 5 residues) to register with these NMR approaches. However, we provide biochemical data in support of a role for D97 and R234 mutations in fundamentally altering more than just structural dynamics at the FnIII-1 loop. Our SEC-MALS data clearly show that D97A and D97K mutations lead to monomerization of RbmA, similar to that observed by truncating the protein in the native-like RbmA* mutant that lacks the FnIII-1 domain loop. By contrast, the R234A mutant maintains dimer formation similar to WT and is even more resistant to proteolysis, suggesting that it locks the protein down into a more ‘closed’ conformation.

*The authors mention that the two states observed in the X-ray data are in fast exchange in the NMR data but that statement is only marginally supported by the data reported in Figure 1. Figure 1 shows discreet positions for discreet mutants. Shift changes are collinear, which is compatible with two-state fast exchange but does not prove that there are only two states, nor that they are in any way related with what was observed crystallographically. The authors mention that RbmA samples both O and D loop states. This can be confirmed by collecting data at different temperatures, which would change the equilibrium position and the exchange kinetics.*

We agree that our conclusion that the collinear chemical shifts we observe with RbmA mutants are indicative of two-state fast exchange is one of a few possible conclusions. However, we believe this is the most parsimonious interpretation of the data, as the NMR data are supported by existing structural data and biochemical assays that probe the two discrete conformational states of the proteins.

Collectively, we believe that the following data provide strong support for our conclusion:

First, the loop in question from the FnIII-1 domain exists in one of these two discrete states (open or closed) in all known crystal structures, several of which originated from different space groups (PDBs: 4BE5, 4BE6, 4BEI, 4KKP, 4KKQ, 4KKR). Therefore, the two states of this loop observed in these crystal structures are unlikely to be an artifact of crystal packing.

Second, the two sets of mutations that we identify in our study appear to recapitulate the two endpoint states of this equilibrium (R234A, closed and D97A/K, open), while the WT protein populates each state approximately equally. We have biochemical data in support of our conclusion that the D97A/K mutants represent the open conformation (i.e., one endpoint of this conformational equilibrium) – namely, we collected NMR spectra on the truncated RbmA* protein that must represent the open state due to truncation of the N-terminal domain – this mutant *cannot* form the closed state because it lacks the required loop in the FnIII-1 domain. *Importantly, spectra of this RbmA* protein directly overlay the D97A/K mutants at each of the sites in the FnIII-2 domain that report on this two-state exchange*, strongly suggesting that the collinear shifts report on fast exchange between an open state and a state defined by the R234A mutant.

Third, based on the known binary conformations of the FnIII-1 domain loop, we propose that it is reasonable to assume that the R234A mutant represents the closed state. We tested this by limited proteolysis, using our series of mutant proteins to explore differences in their conformational states. Our data clearly demonstrate that the D97A mutant is highly sensitive to proteolysis at the interdomain linker and is rapidly cleaved to the minimal FnIII-2 domain that is similar to RbmA*. This same proteolytic fragment is also observed in the wild-type protein although at much lower abundance, consistent with fast exchange of this open conformation with a more closed and protected conformation. By contrast, the R234A mutant is largely resistant to proteolysis, consistent with a more stable, closed conformation. As the reviewer points out, it is true that the NMR data in “Figure 1 shows discreet positions for discreet mutants”; however, it is quite rare to have three reporter sites, all quite distant from the site of mutation (see Figure 1—figure supplement 2, panel G), that exhibit collinear shifts that report on the same putative behavior. Therefore, we believe that our biochemical data collectively support our conclusion that the collinear chemical shifts report on a two-state equilibrium in fast exchange between open and closed loops.

We appreciate the reviewer’s suggestion to use temperature as a variable to probe exchange kinetics – this is a great idea. However, we have not been able to collect usable NMR data on ^[15]^N RbmA at significantly different temperatures, as we are limited by the size and stability of the protein. In order to get decent TROSY spectra on the full-length proteins (a ~60 kDa dimer for R234A and WT proteins), we already must collect data at 35°C – in fact, spectra obtained at 25°C exhibit significantly poorer quality (data not shown). However, the protein is not stable at temperatures above 40°C for the prolonged times required for acquire these TROSY spectra (~3 hrs) at the low concentrations of RbmA that we had to work with. Therefore, we have not yet been able to use temperature to probe dynamics in this system.

*3) Clarify more explicitly in the Introduction what the question is that this manuscript seeks to address. Explain clearly what the novelty is. Explain how your study extends the work of Giglio 2013. If this is the first study to show at the molecular level how higher-order structure assemble in the matrix, state that. If not, explain how is it builds on other efforts of this type. In the Discussion, remind the reader of what makes this work particularly significant. Comment on how generalizable the results might be.*

We have modified the Introduction and Discussion to better address the objective and novelty of our study.

*4) Fix figures so they can be read by a color-blind person. Helpful guidelines can be found here: http://www.nature.com/nmeth/journal/v8/n6/full/nmeth.1618.html*

We apologize for this oversight and have eliminated red-green combinations in all superimposed spectra. We have also check that all figures are readable by a color-blind person.